

# Characterization of Canadian High Arctic glacier surface albedo from MODIS C6 data, 2001-2016

Colleen A. Mortimer[1] and Martin Sharp[1]

[1]Department of Earth and Atmospheric Sciences, University of Alberta, Edmonton, T6G 2E3, Canada

*Correspondence to*: Colleen A. Mortimer (cmortime@ualberta.ca)

**Abstract.** Inter-annual variations and longer-term trends in the annual mass balance of glaciers in Canada's Queen Elizabeth Islands (QEI) are largely attributable to changes in summer melt. The largest source of melt energy in the QEI in summer is net shortwave radiation, which is modulated by changes in glacier surface albedo. We used measurements from the Moderate Resolution Imaging Spectroradiometer (MODIS) sensors to investigate large scale spatial patterns and temporal

trends and variability in the summer surface albedo of QEI glaciers and their relationship to observed changes in glacier surface temperature from 2001 to 2016. Mean summer black-sky shortwave broadband albedo (BSA) decreased at a rate of $0.029 \pm 0.025$ decade[-1] over that period. Larger reductions in BSA occurred in July ($-0.050 \pm 0.031$ decade[-1]). No change in BSA was observed in either June or August. Most of the decrease in BSA, which was greatest at lower elevations around the margins of the ice masses, occurred between 2007 and 2012 when mean summer BSA was anomalously low. The First

Principal Component of the 16-year record of mean summer BSA was well correlated with the mean summer North Atlantic Oscillation Index, except in 2006, 2010, and 2016. During this 16-year period, the mean summer LST increased by $0.046 \pm 0.036$ °C yr[-1] and the BSA record was negatively correlated ($-0.64$, p<0.01) with the LST record, indicative of a positive ice-albedo feedback that would increase rates of mass loss from the QEI glaciers.

## 1 Introduction

The area of glaciers and ice caps in the Queen Elizabeth Islands (QEI, Fig. 1), Arctic Canada, in 2000 was ~104,000 km[2] (Arendt et al., 2013). From 2000 to 2015, the summer mean surface temperature of glaciers in this region increased at a rate of $0.06 \pm 0.04$°C yr[-1], and summer mean air temperatures from 2005-2012 were > 1.0°C warmer than the 1948-2015 mean (Mortimer et al., 2016). QEI summer mean air and glacier surface temperatures are strongly correlated with the annual and summer glacier mass balances, which have become increasingly negative since at least 2003 (Gardner et al., 2013; Lenaerts

et al., 2013; Wolken et al., 2016). Inter-annual variations and longer-term trends in annual glacier mass balance in the QEI are dominated by changes in summer melt (Koerner, 2005), and net shortwave radiation is the largest source of melt energy on the QEI ice caps (Gascon et al., 2013). Variability in net shortwave radiation is strongly modulated by changes in the surface albedo (van den Broeke et al., 2011; Tedesco et al., 2016), the ratio of reflected to incoming solar radiation.





The high albedo of fresh snow declines naturally over time due to settling and grain growth (Warren, 1982). This initial decrease in albedo raises the shortwave energy absorption, leading to warming and/or melt, and a further lowering of the surface albedo. Warmer temperatures and increased snowpack water content further accelerate grain growth, causing more rapid albedo decline that enhances surface warming and/or melt (Wiscombe and Warren, 1980; Colbeck, 1982). A positive snow/ice albedo feedback has been linked to accelerating high-latitude warming, and is increasingly recognized as an important factor in explaining recent increases in rates of mass loss from the Greenland Ice Sheet (Tedesco et al., 2016). On glaciers and ice caps, surface warming and increased melt in summer can lead to earlier and more widespread removal of the previous winter's snowpack, and earlier and more prolonged exposure of underlying low-albedo glacier ice. Albedo decreases can also be caused by aerosol deposition (Warren and Wiscombe, 1980), enhanced biological activity on glacier surfaces (Fountain et al., 2004), and accelerated release of impurities from melting snow and ice, which become concentrated at the snow/ice surface (Clarke and Noone, 1985; Conway et al., 1996; Flanner et al., 2007; Doherty et al., 2010). Given the observed increases in air and glacier surface temperatures across the QEI (Mortimer et al., 2016) we anticipate a reduction in the surface albedo in this region, unless warming has also been accompanied by an increase in solid precipitation that is large enough to raise the surface albedo. The expected albedo reduction has, however, yet to be documented or quantified.

Decadal-scale declines in the surface albedo of the Greenland Ice Sheet, which lies immediately to the east of the QEI, have been simulated using regional climate models, and documented using remote sensing data validated by in situ measurements (e.g. Stroeve et al., 2005, 2013; Casey et al. 2017; Box et al., 2012; 2017; Alexander et al., 2014; Tedesco et al., 2016). Global assessments of land surface albedo (e.g. He et al., 2014) have included the QEI, but these analyses were extremely broad in nature and the detailed spatial pattern of glacier albedo change and its variation over time are unknown. Between 2000 and 2015, measured increases in summer air and glacier surface temperatures in the QEI were greatest in the north and west of the region (Sharp et al., 2011; Mortimer et al., 2016) but we do not know whether there is a similar spatial pattern in the albedo record. Field-based measurements provide information about the surface albedo at specific locations, but there are no long-term spatially distributed in-situ records of the surface radiation budget of glaciers and ice caps in the QEI.

Remote sensing methods enable monitoring of the surface albedo and its spatial and temporal variability at the scale of both individual ice caps and the regional glacier cover. Here, we use measurements of surface albedo from the Moderate Resolution Imaging Spectroradiometer (MODIS) sensors to present the first near-complete picture of variations in the summer surface albedo of QEI glaciers and ice caps from 2001 to 2016. We characterize the spatial and temporal variability in summer albedo, quantify the rate of albedo change across the QEI, and investigate spatial patterns in the relationship between the mean summer albedo and mean summer land surface temperature (LST) records over the QEI ice caps and glaciers from 2001 to 2016.



## 2 Data and methods

Observations from the Moderate Resolution Imaging Spectroradiometers (MODIS), aboard the Terra (2000 to present) and Aqua (2002 to present) satellites, are used to assess the spatial and temporal evolution of the surface albedo over the QEI glaciers and ice caps in summer (June-August). We use the MODIS/Terra and Aqua BRDF/Albedo Daily L3 Global – 500m Collection 06 (C6) product (MCD43A3, Schaaf and Wang (2015)), which provides both white-sky (bi-hemispherical reflectance under isotropic conditions) and black-sky (directional hemispherical reflectance) shortwave broadband surface

albedo (Schaaf et al., 2011; https://www.umb.edu/spectralmass/terra_aqua_modis/v006, and references therein). MCD43A3 albedo is calculated daily for local solar noon from atmospherically corrected surface reflectance measurements from sensors on both the Terra and Aqua satellites over a 16-day period that is centered on the ninth day of each 16-day moving window (https://www.umb.edu/spectralmass/terra_aqua_modis/v006). A semi-empirical Bidirectional Reflectance Distribution Function (BRDF) model, which describes the surface scattering/reflectance of a target as a function of illumination, is used

to estimate surface albedo from directional surface reflectance information recorded by the MODIS sensors (Schaaf et al., 2002, 2011; Jin et al., 2003; Salomon et al., 2006). MCD43A3 white- and black-sky albedos are estimated from Level 2G-lite surface reflectances (MOD/MYD 09) for seven visible and near-infrared bands (spanning 0.4 to 2.4 µm) and three broad bands (shortwave (0.3-5.0 µm), visible (0.3-0.7 µm) and near infrared (0.7-5.0 µm)) in one of two ways. If sufficient (>7) multi-date cloud-free observations with angular sampling sufficient to fully characterize the viewing/illumination geometry

are acquired during a 16 day period, a high quality *full inversion* is run using a semi-empirical RossThick LiSparse Reciprocal (RTSLR) kernel-driven BRDF model (Wanner et al., 1997; Lutch et al., 2000; Schaaf et al., 2002, 201b). If insufficient observations (<7) are available, then a lower quality *magnitude inversion*, which relies on a priori knowledge to scale an archetypal BRDF, is used to estimate the surface albedo (Strugnell and Lutch, 2001; Schaaf et al., 2002; Jin et al., 2003; Liu et al., 2009). Data quality flags, provided in the MCD43A2 data quality assessment product, indicate whether

albedo values [for each pixel] were obtained using the *full* or *magnitude inversion*. Since this study aims to generate an initial assessment of the spatio-temporal variability of the surface albedo of glaciers and ice caps in the QEI with good spatial coverage, both *full* and *magnitude inversion* data are used. Although *magnitude inversions* produce lower quality albedo estimates than the *full inversion* method, previous work using the C5 data showed that the *magnitude inversion* data provide a good representation of the seasonal and spatial patterns in surface albedo (Schaaf et al., 2011; Stroeve et al., 2013).

To our knowledge, no recent research comparing the *magnitude* and *full inversion* retrievals has been published for the MCD43A3 C6 data. Comparison of the MODIS C5 and C6 *full inversion* albedo data from the Greenland Ice Sheet confirmed many of the broad spatial patterns in surface albedo identified in the C5 data, but the magnitude of the C6 albedo change was much smaller (Casey et al., 2017).

The principal uncertainties in MODIS-derived surface albedo data arise from cloud contamination and sensor degradation.

Similarity in the spectral signatures of snow, ice, and thin cloud makes it difficult to discriminate between these surface types (Strabala et al., 1994; King et al., 2004), and the conservative MODIS cloud mask tends to detect more clouds than



actually occur over snow and ice (Ackerman et al., 1998; Hall et al., 2008a). Thus, the absence of observations for periods when clouds are present and the removal of data for periods when clouds are detected may introduce variability in the albedo record that is not representative of true physical change. Despite this, the MCD43A3 albedo product has been found to

provide a reasonable representation of the seasonal albedo cycle over glaciers and ice caps (e.g. Stroeve et al., 2006). Hence, in the absence of long-term ground measurements of glacier surface albedo in the QEI, we made the assumption that this is also the case in that region.

The MODIS sensors are currently operating well beyond their expected [productive] six year lifetimes and some of the instruments are degrading (Wang et al., 2012; Toller et al., 2013). Systematic (decreasing) temporal trends are present in

measurements in the visible and NIR (bands 1-7) of the MODIS C5 data (Lyapustin et al., 2014). Calibration degradation effects, which are largely confined to the Terra sensor, are greatest in the blue band (B3) and decrease with increasing wavelength (Lyapustin et al., 2014). Over time, un-corrected sensor degradation gives rise to decreasing measured surface radiances, which may result in apparent MODIS-derived albedo declines that differ from the true physical change. Contrary to the findings of earlier work that identified strong albedo declines over the dry snow zone of the Greenland Ice Sheet from

2001-2013 using the C5 data (e.g. Stroeve et al., 2005, 2013; Box et al., 2012; Alexander et al., 2014), a recent investigation concluded that much of the observed decline in broadband albedo from Terra-only data resulted from sensor degradation (Polashenski et al., 2015).

Long-term drifts in sensor calibration are addressed in the C6 data used in this study. A new calibration technique that uses in-orbit data from pseudo time-invariant desert site targets is in the calculation of C6 surface reflectances (Toller et al.,

2013). This vicarious calibration approach has improved the precision of C6 reflectances, and it also mitigates long-term sensor drift, particularly at the shorter wavelengths. The C6 post-calibration residual error is on the order of several tenths of one percent for top of the atmosphere (TOA) reflectance (Lyapustin et al., 2014), but larger discrepancies between calibrated Aqua and Terra measurements have been observed in the most recent data (post 2014) (Casey et al., 2017). MODIS C6 data have been shown to be capable of identifying trends in surface albedo >0.01 decade$^{-1}$ (Lyapustin et al., 2014). A recent

analysis of summer (June-August) ice surface albedo changes from MODIS C6 data identified statistically significant declines in surface albedo over the wet snow zone of the Greenland Ice Sheet during the period 2001-2016. For the most part, these decreases in surface albedo are thought to be physically real (Casey et al., 2017).

There are no long-term, spatially distributed, in situ albedo records from the glaciers and ice caps in the QEI, so a comparison between the MCD43A3 records and ground observations is not possible, and this is a limitation of this study.

Ground-truthing of the MOD10A1 C6 albedo product over the Greenland Ice Sheet, immediately adjacent to the QEI, has been undertaken using in situ measurements from the Greenland Climate Network (GC-Net) and the Programme for Monitoring of the Greenland Ice Sheet (PROMICE) (Box et al., 2017). This study found the MOD10A1 C6 data to be a reasonable representation of the true surface albedo. In the absence of quality spatially distributed field measurements of surface albedo from our study area, we assume that this is also the case on the QEI ice caps.




## 2.1 MCD43A3 data processing

Summer (1-2 June (day 152) to 30-31 August (day 243)) MODIS MCD43A3 and MCD43A2 Collection 6 data for MODIS tiles h17v00, h16v00, h16v01, and h15v01 for the period 2001-2016 were obtained from the NASA/USGS Land Processes Distributed Active Archive Center (Schaaf and Wang (2015), http://lpdaac.usgs.gov/ accessed November 2016). Daytime
clear-sky white- and black-sky shortwave broadband and visible albedo data (MCD43A3) and accompanying quality assessment information (MCD43A2) were extracted from the hierarchical data format files and re-projected from the standard MODIS sinusoidal projection to a North America Albers Equal Area projection, WGS84 datum, 500 m resolution, using the MODIS re-projection tool version 4.1 (https://lpdaac.usgs.gov/tools/modis_reprojection_tool). The maximum summer (June-August) solar zenith angle over our study area (74°) was below the product's stated accuracy (<75°, Vermote
et al., 2011; Wang et al., 2012), so no additional filtering was performed to remove data with high solar zenith angles.
The white- and black sky albedos (representing completely diffuse and completely direct illumination, respectively) represent extreme estimates of the actual (blue-sky) bi-hemispheric surface albedo. To avoid redundancy, only results for the black-sky albedo (BSA) (which are fully consistent with those obtained using the white-sky albedo (WSA)) are presented here. The BSA was selected because our analysis focuses on albedo retrieved under clear-sky conditions. This approach is
consistent with previous work using MCD43A3 data (e.g. Alexander et al., 2014; Tedesco et al., 2016; Casey et al., 2017). Unless otherwise specified, henceforth BSA refers to the shortwave broadband black-sky albedo.

## 2.2 MODIS LST (MOD11A2)

The Eight-Day L3 Global Land Surface Temperature and Emissivity product (MOD11A2) C5, which has been found to be a reasonable proxy for the duration and/or intensity of summer melting in the QEI (Sharp et al., 2011; Mortimer et al., 2016),
was used to investigate the relationship between surface temperature and albedo. MOD11A2 daytime and night-time LSTs are computed from MODIS channels 31 (11 µm) and 32 (12 µm) using a split-window technique and all available daytime clear-sky scenes from the Terra satellite for sequential eight day periods (Wan et al., 2002). These data have a spatial resolution of 1 km and nominal product accuracy of ± 1°C but the accuracy over snow and ice surfaces can be as low as ±2°C (Hall et al., 2008b; Koenig and Hall, 2010). Pixels for which the average LST error (QC_Day LST error flag) exceeded
2°C were removed from the analysis and any remaining pixels having a temperature >0°C were assigned a temperature of 0°C (e.g. Hall et al., 2008a; Mortimer et al., 2016). Uncertainties in the MOD11A2 LSTs arise mainly from cloud contamination (Box et al., 2012; Hall et al., 2012) and the removal of observations when clouds are detected (Ackerman et al., 1998; Hall et al., 2008a). Variability in the number of clear-sky days within each observation period and from one year to the next was not found to introduce significant variability in the MODIS-derived LST relative to the true surface temperature
in the QEI (see Mortimer et al., 2016). MOD11A2 data were downloaded from (https://lpdaac.usgs.gov/, accessed September 2014 - October 2015 and June 2017) and re-projected to a North America Albers Equal Area projection, WGS84 datum, 1 km resolution.





### 2.3 Mean summer BSA and LST

Annual precipitation in the QEI is low (<400 mm yr$^{-1}$) and varies little from one year to the next. In contrast, the annual
temperature range is large (> 40°C) (Braithwaite, 2005). Inter-annual variability in QEI annual mass balance is dominated by
changes in the summer mass balance (Koerner, 2005), which, in turn is strongly correlated with summer air temperature
(Sharp et al., 2011). Spatial and temporal patterns in BSA and LST were, therefore, evaluated for the summer months (June-
August). For each year during the 2001-2016 period, mean summer (JJA) BSA was calculated for pixels having at least 10
BSA observations in each month (June, July, August) and at least 45 of a possible 92 observations during the JJA period.
These monthly thresholds ensure both an even distribution of BSA data throughout the summer season and consistency
between different years. Mean summer LST was calculated following the methods of Mortimer et al. (2016) where the mean
summer LST is calculated for pixels having at least 7 of a possible 12 observations between 1-2 June (day 153) and 28-29
August (day 241).

Mean summer (JJA) BSA and LST and mean monthly (June, July, August) BSA anomalies were calculated on a pixel-by-
pixel basis relative to the 2001-2016 mean for pixels having mean summer observations in 11 or more years. This constituted
~87% and ~93% of possible BSA and LST pixels, respectively. Long-term rates of change in BSA and LST over the period
2001-2016 were determined by linear regression between the 16 year records of mean summer LST and BSA and time.
Consistent with the BSA and LST anomalies, regressions were computed on a pixel-by-pixel basis for all pixels having mean
summer observations for 11 or more years. Following Casey et al. (2017), albedo trends below the detection limit of the C6
data (0.01 decade$^{-1}$ (Lyapustin et al., 2014)) are considered to be negligible. To explore whether there were any other spatial
patterns that differed from the long-term (linear) trend, a Principal Components Analysis of the 16 year mean summer BSA
record was performed using data from all pixels with mean summer BSA observations in every year (50% of pixels).

To investigate the spatial pattern of the relationship between surface temperature (LST) and albedo (BSA), linear
correlations between the 16 year LST and BSA records were computed. The MCD43A3 C6 albedo data are produced daily
whereas the MOD11A2 LST data are produced only every eight days. For this direct comparison between the LST and BSA
data, eight day BSA averages were computed from the daily data for the same eight day periods as the MOD11A2 LST
product, and resampled to a 1 km spatial resolution (nearest neighbour resampling). For each year, mean summer BSAs were
computed from these eight day averages for all pixels having at least seven of a possible twelve observations, consistent with
the computation of mean summer LST (Sect. 2.2). The difference between the mean summer BSA values derived from these
8-day averages and those computed from the daily data (0.008) is within the uncertainty of MODIS reflectance products
(0.05 for solar zenith angle <75°; Vermote et al., (2011)). Linear correlations between the 16 year BSA (eight day averaged)
and LST records were then computed on a pixel-by-pixel basis for all pixels having LST and BSA observations in all years
(~80% of all possible pixels).

To ensure that only data for glaciated surfaces were retained, all BSA and LST outputs used in this analysis were clipped to
the Randolph Glacier Inventory v3.2 region 32 (Arctic Canada North) reference polygons (Arendt et al., 2013; Pfeffer et al.,



2014). Surface elevations were obtained from the Canadian Digital Elevation Dataset (CDED) edition 3.0, scale 1:50 k, re-sampled to a 500 m resolution.

## 3 Results

### 3.1 Mean summer albedo

Annual maps of the mean summer clear-sky broadband shortwave black-sky albedo for all glacier-covered surfaces in the QEI for the 2001-2016 period are presented in Fig. 2. The QEI-wide mean summer BSA, averaged across all 16 years, was $0.550 \pm 0.115$ (mean $\pm$ 1 standard deviation). The lowest QEI-wide mean summer BSA ($0.539 \pm 0.127$) was recorded in 2011 while the highest ($0.668 \pm 0.089$) was recorded in 2013 (Table 1). July had the lowest 16-year monthly mean BSA ($0.551 \pm 0.131$), followed closely by August ($0.579 \pm 0.127$) (Table 2, Fig. S1-S3). In each year during the 2001-2016

period, the highest summer monthly BSA was always recorded in June while the lowest monthly BSA was recorded in either July or August. The lowest monthly mean BSA values for June and August were recorded in 2011; for July, the lowest monthly mean occurred in 2012. The highest monthly mean BSA for both June and July occurred in 2013; for August, it occurred in 2003 and may indicate early onset of snowfall that fall.

In general, mean summer BSA is lower around the margins of the ice masses than in the higher elevation interior regions

(Fig. 2). Aggregating the 2001-2016 average mean summer BSA into 50 m elevation bins, we observed a linear rate of BSA increase with elevation (0.0085 per 50 m elevation bin, $r^2 = 0.99$). During years when the QEI-wide mean summer BSA was low (e.g. 2011), we observed a broad zone of low albedo values ($< 0.4$) around the margins of the major ice masses (Fig. 2). Conversely, in years when the mean summer BSA was high (e.g. 2013), this zone was much less obvious. High data dropout at high elevations on Axel Heiberg Island and over the summit of the Devon Ice Cap in 2014 and 2006 (Table S1), may have

produced a negative albedo bias for these regions, since the albedo is typically greater at higher elevations.

### 3.1.1 Albedo anomalies: 2001 to 2016

QEI-wide and regional BSA anomalies, relative to the 2001-2016 mean, were positive from 2001 to 2006 and negative from 2007 to 2012 (Table 1, Fig. 3). Positive BSA anomalies were observed in 2013 and 2014, while 2015 and 2016 saw a return to negative anomalies. For each region shown in Fig. 1, the most negative regionally-averaged BSA anomalies occurred in

either 2001 or 2012, while the most positive regionally-averaged BSA anomalies occurred in either 2004 or 2013 (Table S2). Negative BSA anomalies during 2007-2012, which indicate a larger absorbed fraction of incoming shortwave radiation relative to the 16-year mean, coincide, and are consistent with positive summer air and glacier surface temperature anomalies in the QEI from 2007 to 2015 (Mortimer et al., 2016).

The lowest spatial variability in mean summer BSA anomalies was observed in years when the QEI-wide mean summer

BSA anomaly was either extremely positive (2004 and 2013) or extremely negative (2011 and 2012) (Fig. 3). In 2004 and





2013 (large positive QEI-wide BSA anomalies), large positive BSA anomalies (> 0.1) were observed mainly at lower elevations around the margins of the ice masses, while BSA anomalies were near zero at higher elevations in the interiors of the ice masses. In 2011 and 2012 (large negative QEI-wide BSA anomalies), a similar, but opposite, spatial pattern was observed, with large negative BSA anomalies occurring at low elevations; expect on the Devon Ice Cap in 2012 when BSA

anomalies over most of the ice cap were near zero (> -0.0125).

In years when the QEI-wide BSA anomaly was near zero (between ~-0.0048 and +0.0030; Table 1), BSA anomalies were of opposite sign to those in the rest of the QEI in a region that includes the eastern half of the Devon Ice Cap, the majority of the Manson Icefield, and the southernmost portion of the Prince of Wales Icefield (Fig. 3). In 2005 and 2006 when the QEI-wide BSA anomalies were small and positive (0.0035 and 0.0030, respectively), BSA anomalies in this region were

negative. Conversely, in 2016 when the QEI-wide BSA anomaly was small and negative (-0.0048), large negative BSA anomalies were observed in this region while only weak negative, or even positive, BSA anomalies occurred elsewhere. This eastern maritime region is located close to open water sources in Baffin Bay (Fig. 1). These areas may moderate surface albedo variability through more frequent and/or persistent snowfall and riming events in warm summers when the open water extent is large (Koerner, 1977; 1979; Alt, 1978).

In years when BSA anomalies in maritime regions of the southeastern QEI were opposite to those in the rest of the QEI, BSA anomalies on the western-most part of Axel Heiberg Island tended to be large (< -0.05; Fig. 3) though of the same sign as the QEI-wide BSA anomaly (Table 1). Examination of the 500 hPa geopotential height anomalies for 2001-2016 in NCEP/NCAR reanalysis R1 data (Kalnay et al., 1996; https://www.esrl.noaa.gov/psd) reveals that in years when the mean summer QEI-wide BSA anomaly was strongly negative (e.g. 2001-2004) (positive (e.g. 2007-2012)) a persistent ridge

(trough) was centered over the north and west of the QEI (which includes Axel Heiberg Island) (Fig. 4). These circulation features are likely responsible for the strong BSA anomalies observed over Axel Heiberg Island.

Finally, when the QEI-wide BSA anomaly was small and positive (<0.03, e.g. 2001 and 2003), BSA anomalies on either side of the north-south trending mountain ranges of eastern Ellesmere and Devon Islands were typically of opposite sign. For example, in 2001, positive BSA anomalies occurred on Devon Ice Cap (southeast QEI) and on the eastern side of the north-

south trending mountain range of eastern Ellesmere Island, while negative BSA anomalies occurred in the north and west of the QEI (Fig. 3). We note, however, that there were also instances of instances of positive BSA anomalies at higher elevations in the interior regions of the ice masses on Axel Heiberg Island and northwest Ellesmere Island. In 2003, negative BSA anomalies dominate the south and east of the QEI while BSA anomalies were positive in the northwest of Ellesmere Island and generally negative on Axel Heiberg Island (western QEI). This southeast-northwest spatial pattern is similar to the

pattern of LST change described by Mortimer et al. (2016) where LST change was greatest in the north and west of the QEI.

The spatial pattern of BSA anomalies also varies from month to month (Fig. S4-S6). June BSA anomalies were generally characterized by a southeast-northwest spatial pattern. In years with strong negative (positive) QEI-wide June BSA anomalies, most of the QEI had negative (positive) anomalies, and the largest negative (positive) anomalies were observed at low elevations around the margins of the ice masses in the continental interior (Fig. S4). In years with weak negative QEI-

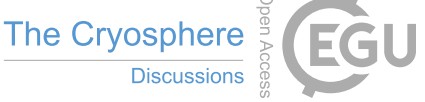

wide June BSA anomalies (e.g. 2002), positive anomalies occurred in the southeast while, negative BSA anomalies occurred in the west and northwest. The opposite scenario occurred in years with weak positive QEI-wide June BSA anomalies (e.g. 2003).

In July, when the mean monthly BSA was typically lowest, BSA anomalies were typically both large (-0.0805 to +0.0843) and less spatially variable than in either June or August, and the sign of the July QEI-wide BSA anomaly was always the

same as that of the mean summer (JJA) QEI-wide BSA anomaly (Table 2, Fig. S5). Mean August BSA anomalies displayed higher spatial variability than July anomalies, but were typically of similar magnitude (-0.0657 to +0.0836). Unlike the June BSA anomalies, which are characterized by a northwest-southeast spatial pattern, spatial variations in the August BSA anomaly are more local (zonal) in nature (Fig. S6). Cyclonic circulation is much more prevalent over the QEI in August than in June and July (Alt, 1987; Gascon et al., 2013). The zonal nature of the August BSA anomaly likely reflects localized

snowfall events related to the passage of individual low-pressure systems, associated with cyclonic circulation conditions. These snowfalls would temporarily raise the surface albedo in affected regions.

**3.2 Albedo change: 2001 to 2016**

Between 2001 and 2016 the mean summer (JJA) QEI-wide BSA decreased at a rate of $0.0029 \pm 0.0025$ yr$^{-1}$ (Table 2, Fig.

54a) and >95% of pixels for which the BSA change exceeded the detection limit of the calibration accuracy experienced a

decrease in mean summer (JJA) BSA. Although the average correlation coefficient of all pixels ($r = 0.31$, $p = 0.24$) was not statistically significant, BSA declines at the $p \leq 0.05$ significance level were observed on all ice masses (Fig. 6a). Furthermore, the measured albedo decline exceeds the detection limit of the MODIS C6 calibration accuracy (0.01 decade$^{-1}$; Lyapustin et al. (2014)), suggesting that BSA did decline somewhat during this 16-year period. There was a larger (but still not significant) decrease in the QEI-wide mean July BSA (-0.0050 $\pm$ 0.0031 yr$^{-1}$, $r = 0.38$, $p = 0.15$), while no QEI-wide

change was observed in either June or August (Table 3, Fig. 5), suggesting that the bulk of the mean summer BSA decline occurred in the month of July. In July, > 93% of pixels exhibited a detectable decrease in BSA and 24% of the measured BSA declines were significant at the $p < 0.05$ level. The BSA change reported here is comparable to, although slightly larger, than that identified for the Greenland Ice Sheet's wet snow zone over the same time period (2001-2016) using similar data (-0.0254 decade$^{-1}$ for MCD43A3 Band 4 (545 – 565 μm)) (Casey et al., 2017).

Between 2001 and 2016, the area-averaged mean summer (JJA) incoming solar radiation over ice covered surfaces in the QEI, computed from daily means of NCEP/NCAR R1 Reanalysis data (Kalnay et al., 1996) (http://www.esrl.noaa.gov/psd/data/gridded/), ranged from 346 W m$^{-2}$ (clear-sky downward solar flux) to 299 W m$^{-2}$ (all-sky downward solar flux). Assuming the solar radiation received at the surface was constant over the 16 year period, our measured BSA reduction (-0.0029 yr$^{-1}$) translates to a total increase in area-averaged absorbed solar radiation of between 1.1

and 1.4 MJ over the 16 year period. For a surface already at the melting point, this equates to an increase in (area-averaged) summer melt of between 0.38 and 0.44 m w.e. (Cuffey and Paterson, 2010 Table 5.1).





Spatially, large reductions in the mean summer BSA ($< -0.005$ yr$^{-1}$) occurred at lower elevations around the margins of the ice masses (where the mean summer BSA is lowest), especially on Axel Heiberg Island, northern Manson Icefield, and on the continental (western) side of ice masses on Ellesmere Island (Fig. 5a). Many of these BSA declines were significant at

the $p \leq 0.05$ level (Fig. 6a). Detectable (although not statistically significant) BSA increases ($> 0.001$ yr$^{-1}$) were observed along the lower portions of outlet glaciers and may point to an increased occurrence of summer snowfall events and/or a later removal of the previous year's snowpack (either from higher fall/winter snow accumulation and/or later melt onset). In addition, in areas close to the oceans, coastal fog may play a role in reducing summer melt over the lower portions of outlet glaciers (Alt, 1987). These variations likely contributed to the low correlation coefficients along the lower reaches of many

outlet glaciers. No detectable change in mean summer BSA was observed over the high elevation ($> \sim$1500 m a.s.l.; $> \sim$1200 m a.s.l for Sydkap Ice Cap) interior regions of the Devon Ice Cap, Sydkap Ice Cap, Agassiz Ice Cap, Prince of Wales Icefield, or northern Ellesmere Island ice caps, even though strong increases in LST were observed in these locations between 2000 and 2015 (Mortimer et al., 2016). Figure 6a also shows a large cluster of pixels with significant declines in mean summer BSA ($\sim -0.00357$ yr$^{-1}$, $p \leq 0.05$) in the interior of the Devon Ice Cap.

The spatial pattern of BSA change in July (the month when BSA decrease was largest) is similar to the spatial pattern of mean summer BSA, with the largest BSA declines occurring at lower elevations (Fig. 5c). In July, BSA trends at the $p \leq 0.05$ significance level (Fig. 6c) coincide, in general, with those locations having the largest BSA declines ($< -0.0075$ yr$^{-1}$). Statistically significant BSA declines occurred in the continental interiors of all the ice masses on Ellesmere Island, as well as on the northern half of Axel Heiberg Island, the southern Prince of Wales Icefield, northern Manson Ice Cap, and

southwest Devon Ice Cap.

Although no significant QEI-wide BSA change was observed in either June ($-0.0017 \pm 0.0024$ yr$^{-1}$) or August ($-0.0022 \pm 0.0036$ yr$^{-1}$), owing to the large amount of spatial variability in the sign and magnitude of BSA change in these months, local areas of consistent BSA change are observed. In June, increases in BSA occurred on the eastern portions of Prince of Wales Icefield and Agassiz Ice Cap, while weak-to-moderate BSA declines ($> \sim -0.005$ yr$^{-1}$) occurred elsewhere. We note, however,

that only 10% of pixels had June mean BSA trends significant at the $p \leq 0.05$ level. These pixels are mainly found on the southwest parts of the Devon and Agassiz Ice Caps and in northwest Ellesmere Island (Fig. 6b).

In August, large and statistically significant BSA declines ($< -0.00625$ yr$^{-1}$, $p \leq 0.05$) occurred on the eastern Devon Ice Cap, Manson Icefield, and southeast Prince of Wales Icefield (Fig. 6d). Moderate August mean BSA declines ($< -0.005$ yr$^{-1}$, $p \leq 0.05$) occurred on the northern half of northwest Ellesmere Island. August mean BSA increased on the summits of Devon Ice

Cap and Sydkap Ice Cap, in the western part of Prince of Wales Icefield, in some eastern and southerly sections of the Agassiz Ice Cap, and on some outlet glaciers in northwest Ellesmere Island and Axel Heiberg Island. These increases, however, were not statistically significant.



### 3.3 Principal components analysis

To explore whether there are any other spatial patterns in the 16-year mean summer BSA record that differ from the long-
term (linear) trend, we performed a Principal Components Analysis of the BSA record (Sect. 2.3). The first and second
Principal Components (Fig. 7) explain 65% and 12% of the variance in the mean summer BSA record, respectively. The
spatial pattern of the First Principal Component (PC1) scores (Fig. 7a) is generally consistent with the spatial patterns of
summer (JJA) and July mean BSA change described previously (Sect. 3.2, Fig. 5a,c). PC1 scores are strongly negative (< -
0.005) on western Axel Heiberg Island (large BSA declines), and weakly negative (> -0.001) or even positive at high
elevations in the interior of the ice masses where no detectable change in mean summer BSA was observed. Moderately
negative (~ -0.002) component scores occur on the western (continental) side of eastern Ellesmere Island's ice masses as
well as over much of Manson Icefield and the southeast portion of Prince of Wales Icefield, where large decreases were
observed in both summer (JJA) and July BSA.

For PC1, the highest Empirical Orthogonal Functions (EOFs) (32.8 and 25.9) correspond to the years with the lowest mean
summer BSA (2011 and 2012) (Fig. 8). The lowest EOFs (-31.7 and -24.6) correspond to the years with the highest mean
summer BSA (2013 and 2004). In addition, the departure from zero is much larger for the minimum component scores than
for the maximum component scores (Fig. 7a), suggesting that both positive and negative BSA anomalies are likely caused by
forcings with the same spatial pattern, albeit with the opposite sign. Investigating possible relationships between surface
albedo and known large-scale patterns of atmospheric variability (Arctic Oscillation, Pacific North American Pattern, North
Atlantic Oscillation), we found the EOFs for PC1 to be well (negatively) correlated with the mean summer North Atlantic
Oscillation (NAO) index ($r = -0.84$, $p < 0.001$, Fig. 8), derived by averaging the June-August monthly mean NAO indices for
2001 to 2016 (http:www.cpc.ncep.noaa.gov). From 2007-2012, when both the NAO index and BSA anomalies were
negative (Fig. 8), the frequency of anticyclonic circulation over the QEI in summer doubled relative to the 1948-2012 mean
(Bezeau et al., 2015). Clear sky conditions that accompany anticyclonic circulation increase the proportion of incoming
shortwave radiation received at the air-ice interface, providing more energy for melt and driving albedo decline.

The spatial pattern of the second Principal Component (PC2) scores (Fig. 7b) resembles that of the August BSA change (Fig.
5d). Large negative component scores (< -0.004) were observed on eastern Devon Ice Cap and Manson Icefield, where large
August BSA declines (< -0.0075 yr$^{-1}$) occurred. Positive component scores were observed in the in southwest Axel Heiberg
Island and at lower elevations on northern Ellesmere Island's southwestern ice caps, where the mean August BSA increased
(Fig. 5d).

PC2 had the largest EOFs in 2006, 2010, and 2016. In these years, the mean August BSA anomaly in these regions was of
opposite sign to that of the mean summer (JJA) anomaly, and there was poor correspondence between the PC1 EOFs and the
JJA NAO index (Fig. 8). Examination of the cumulative mass change record for the QEI from the Gravity Recovery and
Climate Experiment (GRACE; Wolken et al. (2016), extended to 2016) shows that in the same years (2006, 2010, 2016),
once the annual minimum glacier mass was reached, there was a prolonged period of constant but low mass (i.e. no melt or




snowfall) before fall/winter accumulation began (inferred from an increase in mass). In other years, there was a sharp transition from the local end-of-summer mass minimum to the period of seasonally increasing mass. There is also good correspondence between the spatial pattern of PC2 scores and the spatial pattern of snow accumulation and $\partial^{18}O$ values that was inferred for the 1962-1974 period from surface snow samples and shallow firn cores collected in spring 1974 (Koerner,

1979). Areas having large negative PC2 component scores (< -0.004) were characterized by relatively high accumulation rates (Fig. 2 and 3 in Koerner (1979)) and snow that is isotopically warm (Fig. 7 in Koerner (1979)). Areas with positive PC2 component scores had lower snow accumulation rates. The correspondence between the pattern of PC2 component scores (Fig. 6b) to previously reported snow accumulation patterns over the QEI, and the spatial pattern of change in the August mean BSA (Fig. 4d), suggests that anomalously low snow accumulation in August may have influenced the mean

summer BSA in those years when PC2 had the largest EOFs (e.g. 2006, 2010, and 2016).

## 4 Discussion

### 4.1 Summer and monthly BSA

Inter-annual variations in the QEI mean summer LST anomalies have previously been linked to variations in the NAO index (Mortimer et al., 2016). Here we find that the 16-year record of summer BSA is also strongly tied to the summer NAO index

(Sect. 3.3). In years with strong negative (positive) BSA anomalies and positive (negative) LST anomalies (e.g. 2007-2012; 2013; see Mortimer et al. (2016)) the NAO index was negative (positive). During the 2007-2012 period when the NAO index was negative (Fig. 8), there was an increase in the frequency of anticyclonic circulation, compared to the 1948-2012 mean (Bezeau et al., 2015). Similarly, over the Greenland Ice Sheet, strong anticyclonic ridging, associated with clear skies and the advection of warm air from the south, was found to co-vary with the NAO index over the period since ~2001

(Rajewicz and Marshall, 2014). These conditions were found to enhance the strength of the ice-albedo feedback on the Greenland Ice Sheet during 2009-2011, resulting in higher rates of melt and glacier mass loss (Box et al., 2012). Although a similar phenomenon is likely to have occurred in the QEI, we find that in some years (2006, 2010, 2016) there was poor correspondence between the mean summer BSA record and the NAO (Fig. 8), suggesting that an additional forcing may be influencing the spatial and temporal variability of glacier surface albedo in the QEI. In 2006, 2010, and 2016, the spatial

pattern of BSA anomalies closely resembles that of the mean August BSA change, which, in turn, closely resembles the spatial pattern of PC2 component scores. In these years, delayed snowfall onset and limited melt in August were inferred from the GRACE mass change record. This could indicate that in some years during the 2001-2016 period, variability in August snowfall, in addition to that tied to the NAO, may have influenced the mean summer BSA.

Spatial variations in the monthly BSA reflect the different atmospheric circulation patterns that occur over the QEI during

the course of the summer. In the QEI, anticyclonic circulation tends to dominate in the months of June and July, while cyclonic circulation often occurs in August (Alt, 1987; Gascon et al., 2013). Investigation of the surface energy balance of



the Devon Ice Cap between 2007 and 2010 found increases in summer (JJA) melt energy were associated with an increase in
net shortwave radiation in June and July, and with an increase in net longwave radiation in August (Gascon et al., 2013).
Strong, persistent anticyclonic circulation maximizes the incoming shortwave radiation received at the air-ice interface, and a
lower albedo increases the proportion of incoming solar radiation that is absorbed, providing more energy for warming and
melt. Our results show that the bulk of the decline in mean summer QEI albedo occurred in July (Sect. 3.2), which is
consistent with previous work (e.g. Alt, 1987; Gardner and Sharp, 2007) that found variability in July near-surface air
temperatures to be the primary influence on inter-annual variability in annual QEI mass balance. Variability in July air
temperatures has, in turn, been associated with the variations in the strength, position, and geometry of the July circumpolar
vortex (Gardner and Sharp, 2007). Extreme high melt years in the QEI are associated with the intrusion of a steep ridge at all
levels in the troposphere and the absence of the North American trough, making the QEI thermally homogeneous with
continental North America (Alt, 1987). Between 2001 and 2016, in warm years with low (e.g. 2007-2012) (high (e.g. 2001-
2004)) albedos, there was a persistent ridge (trough) in the 500 hPa geopotential height surface centered over the north and
west of the QEI (Fig. 4; Sect. 3.1.1;). This configuration appears to be tied to the increased warming and albedo declines
observed in the central western part of the QEI from 2007 to 2012. For example, strong warming and albedo declines over
Axel Heiberg Island and north-central Ellesmere Island coincided with a ridge of high pressure centered over the north and
west of the QEI that was often observed in years when the NAO index was negative (Fig. 8).

In contrast to June and July, when the largest BSA declines occurred in the west of the QEI, BSA declines in August were
largest at low elevations in the maritime regions of eastern Devon Ice Cap, Manson Icefield, and the southwestern Prince of
Wales Icefield (Fig. 5). August weather in the QEI tends to be dominated by cyclonic circulation (Alt, 1987; Gascon et al.,
2013). Low pressure systems which track from the southwest to the north and northeast are common in August, and they
advect warm moist air into the Arctic from the south. In the eastern QEI, orographic uplift of air masses tracking from the
southwest, and subsequent adiabatic heating of these air masses when they descend on the eastern sides of ice masses, would
bring warm dry air to the eastern (lee) side of the mountains in the eastern QEI, promoting both warming and albedo decline
in these regions in August.

Differences in the dominant atmospheric circulation patterns over the course of the summer season are responsible, at least
in part, for where and in which month, the largest albedo declines occur. Albedo lowering increases the proportion of
incoming solar radiation absorbed at the air-ice interface, and thus the energy available to drive further melt and surface
albedo decline. Knowing where and when albedo changes are likely to occur in future is, therefore, important for predicting
future rates of mass loss from the QEI ice caps. Recent investigations of atmospheric circulation patterns over the QEI (e.g.
Gardner and Sharp, 2007) focused on characterization of July temperature and atmospheric conditions, since July is usually
the month when melt rates peak. Our results suggest, however, that changes occurring during the month of August are also
important, especially as the length of the melt season continues to increase.

Although large-scale circulation patterns appear to explain much of the broad spatial pattern (and seasonal variability) of
BSA change, the physiography of the QEI introduces considerable local variation into the way these changes are expressed.




Variations in the magnitudes of both inter-annual variability and longer-term changes in annual precipitation, summer melt duration, and melt intensity have previously been associated with distance from moisture sources (Koerner, 1979; Alt, 1987; Wang et al., 2005). Snow accumulation rates in the QEI tend to decrease from east to west, and Baffin Bay is the largest moisture source for the eastern QEI (Koerner, 1979). Summer snowfall events temporarily raise the surface albedo, while

locally higher fall and winter snow accumulation can delay the exposure of bare ice and firn in the following summer relative to areas with lower accumulation rates. However, although BSA decreases were generally larger in the north and west, equally large decreases occurred at low elevations on Manson Icefield and southeast Prince of Wales Icefield in the eastern QEI. These large BSA decreases at low elevations in the eastern QEI may arise from a decoupling of high and low elevation temperatures in low melt years. For example, Alt (1978) found that, in some years when there was very little melt

near the summit of the Devon Ice Cap, the Sverdrup Glacier (lower elevation) experienced very high melt rates. Further, Wang et al. (2005) attributed comparatively high melt rates observed at low elevations on the Prince of Wales Icefield in 2002 to a decoupling of the temperature regimes experienced at high and low elevations, similar to that described by Alt (1978).

**4.2 Relationship between albedo change and temperature**

During the period 2001-2016, mean summer QEI-wide LST (16-yr average: -3.25 ± 1.73 °C) increased at a rate of 0.046 ± 0.036 °C yr$^{-1}$ (Table 3). The spatial pattern of LST increase (Fig. S7b) is opposite to the pattern of mean summer BSA change, in that BSA decreases were largest at lower elevations where the mean summer BSA was lowest (Fig. 5a), while LST increases were greatest at higher elevations where the mean summer LST is lower. This observation may be explained by the fact that, at lower elevations, where the mean summer LST regularly reached the melting point, there was less

potential for warming than at higher elevations (Mortimer et al., 2016). Investigation of the spatial pattern of the relationship between LST and BSA (derived from linear correlations between the 16 year LST and BSA records for each pixel (Sect. 2.3)), shows that, as expected, LST and BSA were negatively correlated over most of the QEI (~99.7% of pixels; Fig. 9). This negative correlation (average correlation coefficient of all pixels: r = -0.64, p < 0.01) points to a positive ice-albedo feedback which would promote enhanced rates of glacier mass loss from the QEI. In general, correlations were stronger in

the north and west of the QEI than in the south and east, with the exception of the northwest Devon Ice Cap (southeast QEI) where large negative correlation coefficients (r < -0.8) were also observed.

The nature of the relationship between LST and BSA varies both within and between regions (Fig. 9). In the eastern QEI [Agassiz Ice Cap, Prince of Wales Icefield, and Devon Ice Cap], correlations tended to be stronger on the western continental side of the north-south running mountain range (where BSA declines were also large) than on the eastern

maritime slopes. There were, however, some exceptions to this pattern, most notably a low elevation region near Cadogan Inlet (Fig. 9a, black box), eastern Prince of Wales Icefield, where large negative correlations (r < -0.8) were observed and BSA and LST changes were both small. The strong negative correlations in the continental interior are likely attributable to



the presence of the north-south trending mountain range in eastern Ellesmere and Devon Islands. In the continental interior
of eastern Devon and Ellesmere Islands, warming of air and glacier surface temperatures resulted in albedo decreases,
leading to further surface warming. Koerner (1979) attributed the existence of a dry area with low amounts of snow
accumulation centered on western Ellesmere Island to the precipitation-shadowing effect of the mountains to the east. In
June and July when circulation is primarily anticyclonic, the presence of a barrier to moisture transport from the east limits
precipitation on the western (lee) side of the eastern ice masses, while adiabatic heating of descending air masses results in
warm dry air which promotes warming and melting, and enhances albedo decline, in the west.

The strength of the relationship between LST and BSA also varies with elevation. Correlations tended to be weaker and
more variable at high elevations, although there are also several instances of low correlations at low elevations, for instance
on southeast Devon Ice Cap and Manson Icefield. At lower elevations, where the mean summer LST is high (Fig. S7a),
weaker correlations between temperature and albedo may reflect the fact that the albedo continues to decline once the
surface temperature has reached the melting point. At high elevations (above ~1500 m a.s.l., Sect. 3.2) BSA decreases were
small or non-existent, and the correlation between LST and BSA was both modest and highly variable (Fig. 9). Moderate
negative correlations ($r > -0.6$) and some instances of positive correlations, were observed at high elevations on Prince of
Wales Icefield and Agassiz Ice Cap. On northwest Ellesmere Island where correlations were generally strongly negative,
high-elevation regions had more moderate correlations and some instances of positive correlations between LST and BSA
were also observed.

LST increases were largest at high elevations (Fig. S7b) and melt occurred at all elevations and at all locations in the QEI at
some point during the 2000-2015 period (Mortimer et al., 2016). Since the rate of grain metamorphism increases with the
temperature and water content of snow, even warmer summer temperatures may be needed to promote large-scale albedo
decline in permanently snow-covered high elevation areas. Alternatively, higher air temperatures in such regions may
promote increased summer snowfall and more frequent riming events, which temporarily raise the surface albedo and limit
long-term albedo decline. We note that in many high elevation regions, where BSA changes were either small or not
observed and correlations between LST and BSA were weak and highly variable, PC1 had positive component scores
(compared with negative scores elsewhere). Although PC1 was strongly correlated with the summer NAO index (Sect. 3.3),
the occurrence of positive component scores in these high elevation regions (compared with large negative scores elsewhere)
may point to an additional mechanism that is affecting the albedo record in these regions. Variability in the extent of open
water in the QEI's inter-island channels has previously been correlated with variability in summer temperatures and 500 hPa
geopotential height anomalies in the QEI (Koerner, 1977; Bezeau et al., 2015). Increased open water extent during warm
years promotes atmospheric convection, which strengthens the advection of warm moist air masses into Baffin Bay
(Koerner, 1979). In the eastern QEI, the high elevation regions of eastern Ellesmere and Devon Islands act as a barrier to this
increased moisture transport (Koerner, 1979). As a result, precipitation is deposited on the eastern maritime slopes, where it
temporarily increases the surface albedo. This scenario provides a plausible explanation for lower rates of BSA decline and





poor correspondence between LST and BSA changes at higher elevations in the eastern QEI. A similar set of processes may also be occurring at high elevations on northern Ellesmere Island where the Arctic Ocean is the primary moisture source.

**5 Conclusions**

This study presents the first complete picture of mean summer surface albedo variations over all glaciated surfaces in the QEI during the period 2001-2016. Mean summer shortwave broadband black-sky albedo decreased at a rate of $0.0029 \pm 0.0025$ yr$^{-1}$ over the 16-year period. Strong negative BSA anomalies from 2008-2012 suggest that the bulk of the observed albedo decline occurred during this five-year period. Large albedo declines occurred in July ($-0.0050 \pm 0.0031$ yr$^{-1}$); while no detectable change in BSA occurred in either June or August, indicating that the bulk of the mean summer BSA decrease is concentrated in July, when strong anticyclonic circulation occurs. The 16-year history of mean summer BSA changes is strongly tied to variations in the summer NAO index, except in the years 2006, 2010, and 2016 when changes in the mean summer BSA appear to be dominated by the effect of changes in the mean August BSA. Albedo declines were largest at low elevations around the margins of the ice masses and the 16-year record of mean summer BSA was negatively correlated with the 16 year record of mean summer LST (which increased at a rate of $0.046 \pm 0.036$ °C yr$^{-1}$ during the 2001-2016 period), indicating the presence of a positive ice-albedo feedback in a warming climate that would enhance rates of glacier mass loss from the QEI.

*Acknowledgements* We thank NSERC Canada (Discovery Grant to MS, Vanier Canada Postgraduate Scholarship to CM), and Alberta Innovates – Technology Futures (MS) for financial support. Scott Williamson provided helpful advice on MODIS data processing. MODIS data are available from https://lpdaac.usgs.gov. NCEP/NCAR R1 Reanalysis data are available from http://www.esrl.noaa.gov/psd/data/gridded/).

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



**Table 1:** Clear-sky BSA and LST; 1 standard deviation. Anomalies are with respect to the 2001-2016 mean.

| Year | BSA mean | BSA mean anomaly* | LST mean | LST mean anomaly* |
|---|---|---|---|---|
| 2001 | 0.631 ± 0.106 | 0.0226 ± 0.0458 | -3.9 ± 1.9 | -0.65 ± 0.63 |
| 2002 | 0.633 ± 0.124 | 0.0316 ± 0.0320 | -3.7 ± 2.1 | -0.43 ± 0.66 |
| 2003 | 0.627 ± 0.109 | 0.0292 ± 0.0329 | -3.9 ± 1.8 | -0.65 ± 0.49 |
| 2004 | 0.655 ± 0.112 | 0.0517 ± 0.0318 | -4.2 ± 2.3 | -1.00 ± 0.73 |
| 2005 | 0.605 ± 0.132 | 0.0035 ± 0.0282 | -2.6 ± 1.7 | 0.67 ± 0.40 |
| 2006 | 0.608 ± 0.120 | 0.0030 ± 0.0282 | -4.2 ± 2.3 | -0.95 ± 0.80 |
| 2007 | 0.586 ± 0.125 | -0.0167 ± 0.0282 | -2.2 ± 1.6 | 1.00 ± 0.44 |
| 2008 | 0.588 ± 0.119 | -0.0117 ± 0.0212 | -2.7 ± 1.4 | 0.54 ± 0.52 |
| 2009 | 0.586 ± 0.115 | -0.0177 ± 0.0220 | -2.8 ± 1.7 | 0.46 ± 0.44 |
| 2010 | 0.601 ± 0.112 | -0.0021 ± 0.0307 | -2.5 ± 1.7 | 0.77 ± 0.49 |
| 2011 | 0.539 ± 0.127 | -0.0651 ± 0.0326 | -2.3 ± 1.3 | 0.90 ± 0.61 |
| 2012 | 0.550 ± 0.126 | -0.0516 ± 0.0309 | -2.4 ± 1.4 | 0.87 ± 0.64 |
| 2013 | 0.668 ± 0.086 | 0.0604 ± 0.0431 | -5.4 ± 2.2 | -2.14 ± 0.85 |
| 2014 | 0.614 ± 0.109 | 0.0153 ± 0.0245 | -3.4 ± 1.8 | -0.18 ± 0.50 |
| 2015 | 0.578 ± 0.124 | -0.0223 ± 0.0248 | -2.8 ± 1.7 | 0.37 ± 0.46 |
| 2016 | 0.600 ± 0.120 | -0.0048 ± 0.0334 | -2.8 ± 1.8 | 0.42 ± 0.68 |
| 2001-16* | 0.599 ± 0.115 | – | -3.2 ± 1.7 | – |

*pixels having mean summer (JJA) BSA observations in at least 11 of a possible 16 years (see Sect. 2.3)





**Table 2:** Clear-sky mean summer monthly BSA^ for the QEI ice cover; ± 1 standard deviation.

| Year | June | July | August | June anomaly* | July anomaly* | August anomaly* |
|------|------|------|--------|---------------|---------------|-----------------|
| 2001 | 0.722 ± 0.084 | 0.584 ± 0.127 | 0.597 ± 0.123 | 0.0359 ± 0.0502 | 0.0285 ± 0.0585 | 0.0153 ± 0.0572 |
| 2002 | 0.680 ± 0.108 | 0.615 ± 0.135 | 0.608 ± 0.145 | -0.0005 ± 0.0424 | 0.0595 ± 0.0435 | 0.0270 ± 0.0466 |
| 2003 | 0.684 ± 0.085 | 0.561 ± 0.139 | 0.658 ± 0.127 | 0.0003 ± 0.0231 | 0.0029 ± 0.0340 | 0.0836 ± 0.0768 |
| 2004 | 0.714 ± 0.085 | 0.620 ± 0.126 | 0.632 ± 0.133 | 0.0317 ± 0.0285 | 0.0638 ± 0.0465 | 0.0501 ± 0.0472 |
| 2005 | 0.676 ± 0.099 | 0.572 ± 0.139 | 0.573 ± 0.143 | -0.0064 ± 0.0300 | 0.0150 ± 0.0364 | -0.0116 ± 0.0468 |
| 2006 | 0.712 ± 0.087 | 0.615 ± 0.129 | 0.568 ± 0.137 | 0.0255 ± 0.0316 | 0.0602 ± 0.0440 | -0.0166 ± 0.0431 |
| 2007 | 0.692 ± 0.090 | 0.544 ± 0.141 | 0.536 ± 0.155 | 0.0099 ± 0.0228 | -0.0143 ± 0.0339 | -0.0518 ± 0.0494 |
| 2008 | 0.660 ± 0.104 | 0.533 ± 0.148 | 0.576 ± 0.118 | -0.0220 ± 0.0298 | -0.0226 ± 0.0353 | -0.0050 ± 0.0345 |
| 2009 | 0.690 ± 0.089 | 0.536 ± 0.132 | 0.533 ± 0.142 | 0.0085 ± 0.0258 | -0.0212 ± 0.0310 | -0.0544 ± 0.0383 |
| 2010 | 0.668 ± 0.091 | 0.528 ± 0.136 | 0.606 ± 0.136 | -0.0143 ± 0.0230 | -0.0289 ± 0.0331 | 0.0188 ± 0.0656 |
| 2011 | 0.628 ± 0.101 | 0.480 ± 0.147 | 0.521 ± 0.148 | -0.0557 ± 0.0347 | -0.0770 ± 0.0400 | -0.0657 ± 0.0517 |
| 2012 | 0.628 ± 0.112 | 0.478 ± 0.146 | 0.559 ± 0.137 | -0.0532 ± 0.0395 | -0.0805 ± 0.0399 | -0.0243 ± 0.0532 |
| 2013 | 0.724 ± 0.072 | 0.640 ± 0.107 | 0.628 ± 0.109 | 0.0402 ± 0.0406 | 0.0843 ± 0.0517 | 0.0466 ± 0.0637 |
| 2014 | 0.698 ± 0.080 | 0.572 ± 0.128 | 0.584 ± 0.136 | 0.0167 ± 0.0277 | 0.0148 ± 0.0335 | 0.0026 ± 0.0495 |
| 2015 | 0.677 ± 0.103 | 0.490 ± 0.143 | 0.587 ± 0.147 | -0.0033 ± 0.0282 | -0.0675 ± 0.0400 | 0.0026 ± 0.0457 |
| 2016 | 0.675 ± 0.089 | 0.545 ± 0.146 | 0.589 ± 0.146 | -0.0066 ± 0.0280 | -0.0146 ± 0.0354 | 0.0049 ± 0.0652 |
| 2001-2016* | 0.680 ± 0.089 | 0.551 ± 0.131 | 0.579 ± 0.127 | – | – | – |

^average of all pixels in each region having at least 11 (10 for June) of a possible 31 (30 for June) observations.

*pixels having June BSA observations in at least 11 of a possible 16 years.







**Table 3**: 2001-2016 BSA and LST change for glaciated regions of the QEI (Fig. 1); ± 1 standard deviation.

| Year | QEI | Agassiz IC | Axel Heiberg I | Devon I & Coburg I | Manson IF | Meighen IC | Northwest Ellesmere I | Prince of Wales IF | Sydkap IC |
|---|---|---|---|---|---|---|---|---|---|
| JJA Mean BSA (yr⁻¹) | -0.0029 ± 0.0025 | -0.0021 ± 0.0022 | -0.0041 ± 0.0031 | -0.0034 ± 0.0021 | -0.0043 ± 0.0026 | -0.0055 ± 0.0025 | -0.0024 ± 0.0025 | -0.0028 ± 0.0021 | -0.0036 ± 0.0030 |
| June Mean BSA (yr⁻¹) | -0.0017 ± 0.0024 | -0.002 ± 0.0023 | -0.0027 ± 0.0027 | -0.0019 ± 0.0016 | -0.0017 ± 0.0028 | -0.0046 ± 0.0017 | -0.0011 ± 0.0026 | -0.0020 ± 0.0024 | -0.0028 ± 0.0022 |
| July Mean BSA (yr⁻¹) | -0.0050 ± 0.0031 | -0.0044 ± 0.0026 | -0.0077 ± 0.0035 | -0.0040 ± 0.0024 | -0.0054 ± 0.0028 | -0.0081 ± 0.0027 | -0.0048 ± 0.0032 | -0.0049 ± 0.0028 | -0.0064 ± 0.0031 |
| August Mean BSA (yr⁻¹) | -0.0022 ± 0.0036 | -0.0010 ± 0.0027 | -0.0024 ± 0.0039 | -0.0043 ± 0.0036 | -0.0057 ± 0.0043 | -0.0050 ± 0.0041 | -0.0014 ± 0.0032 | -0.0016 ± 0.0030 | -0.0018 ± 0.0049 |
| JJA LST (°C yr⁻¹) | 0.05 ± 0.04 | 0.05 ± 0.03 | 0.07 ± 0.04 | 0.04 ± 0.03 | 0.02 ± 0.02 | 0.07 ± 0.06 | 0.06 ± 0.03 | 0.04 ± 0.03 | 0.04 ± 0.03 |













**Figure 1: Glaciated regions of the Queen Elizabeth Islands, Arctic Canada. Base Image: Moderate Resolution Spectroradiometer, 4 July 2011. Top-right inset: red polygon shows location of Queen Elizabeth Islands, Canada. Bottom-left inset: the eight regions used in this study.**




**Figure 2: Mean summer clear-sky shortwave broadband black-sky albedo for the QEI.**






**Figure 3: Mean summer clear-sky shortwave broadband albedo anomaly for the QEI relative to the 2001-2016 mean.**




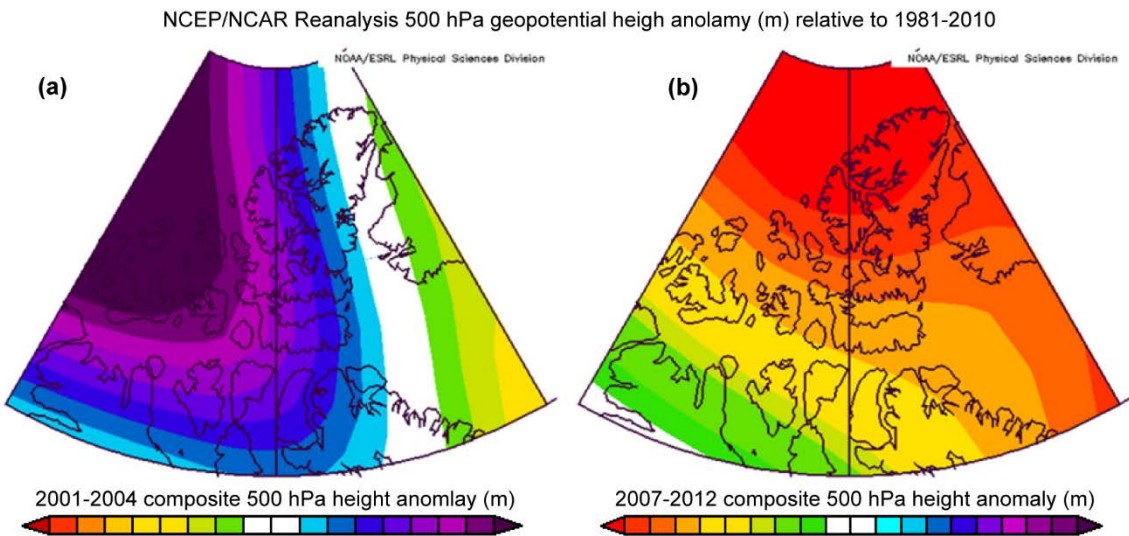

**Figure 4: Mean summer (JJA) composite NCEP/NCAR Reanalysis 500 hPa geopotential height anomaly for (a) a period of large negative BSA anomalies (2001-2004) and (b) a period of large positive BSA anomalies (2007-2012). Source: https://www.esrl.noaa.gov/psd.**





**Figure 5: Linear rate of change (yr⁻¹) in (a) mean summer (JJA), (b) June, (c) July, and (b) August, clear-sky shortwave broadband black-sky albedo for 2001-2016 for the QEI.**



**Figure6: p-value of the linear regression (Fig.4) of (a) mean summer (JJA), (b) June, (c) July, and (b) August, clear-sky shortwave**
**broadband black-sky albedo for the period 2001-2016 for the QEI.**



**Figure 7: Component scores for the first two Principal Components of the mean summer clear-sky BSA (Fig. 2) for the QEI.**





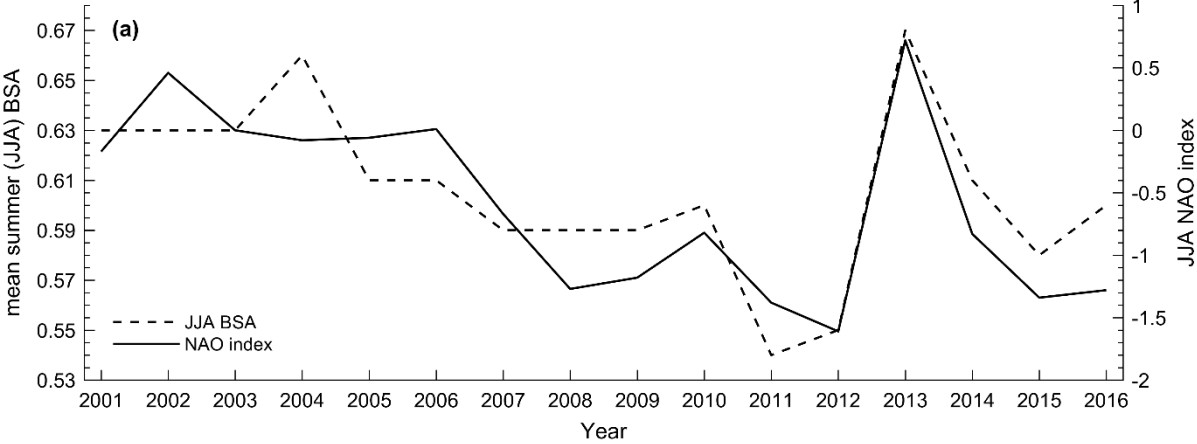

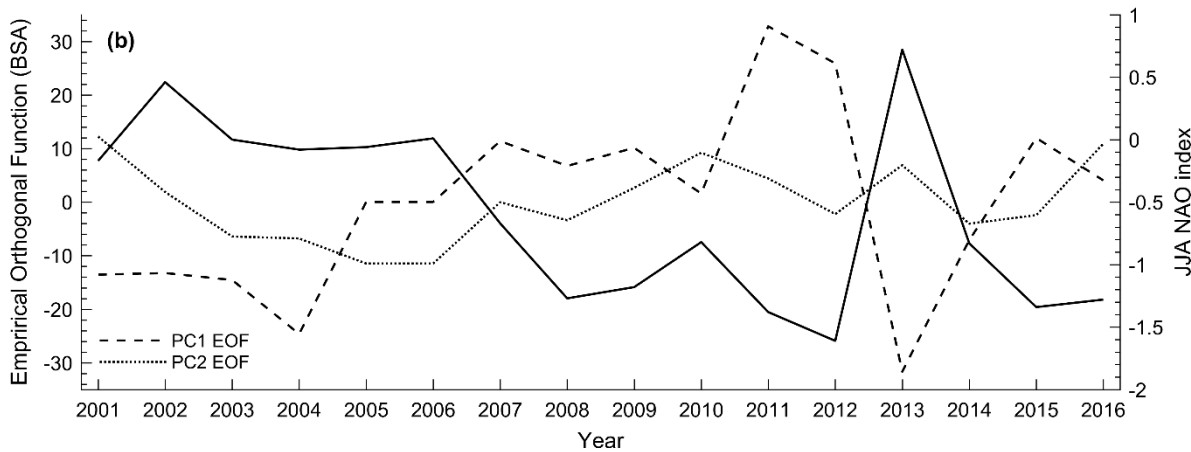


**Figure 8: (a) mean summer (JJA) clear-sky shortwave broadband black-sky albedo (left-hand axis) and the mean summer (JJA) NAO index (right-hand axis) for 2001-2016. (b) Empirical Orthogonal Function (EOF) for the First and Second Principal Components of the 16 year mean summer BSA record (left-hand axis) and the mean summer (JJA) NAO index (right-hand axis) for 2011-2016.**




**(a)**

**(b)**

Figure 9: (a) Pearson Correlation Coefficient (r) and (b) p-value for linear regression of the 16 year BSA and LST record. Black box shows location of Cadogan Inlet (Sect. 4.2).