# Peer review of "Spatiotemporal variability of Canadian High Arctic glacier surface"

_The Cryosphere, 2017_

## Referee Comment (RC1) · Anonymous Referee #1 · 13 Sep 2017

General Comments:

In this paper, the albedo and surface temperature of the glaciers in the Queen Elizabeth Islands (QEI) are investigated using MODIS data products for 2001 – 2016. Over the study period BSA is shown to decrease significantly in the month of July, while LST increases during the summer months (JJA). This is a valuable paper and it is well written (though a bit long) and technically sound. My comments below are almost all optional.

I believe that the authors have not fully taken advantage of this opportunity to explore the relationship between albedo and LST. Though they point out that albedo and LST

[Figure]

are negatively correlated for the QEI, they do not discuss that relationship as cause-and-effect in either the Abstract or the Conclusion. The relationship between albedo and LST is discussed in Section 4.2 but I would have liked that discussion to have been better integrated throughout the paper. Since there was quite a bit of information about LST in the paper, the authors might want to consider including LST in the title of the manuscript.

The authors say that only the month of July shows the statistically-significant albedo decrease over the study period. Thus they should consider stating that the July albedo is decreasing over the study period instead of saying mean summer (JJA) albedo is decreasing.

If the paper can be made more concise it would be easier to follow.

Specific comments:

Title: consider removing "C6" from the title since it's not that different from C5 for the albedo and LST, and many people, before reading the paper, have no idea what C6 is. You have appropriately mentioned the fact that you've used C6 in the paper as needed. Also consider including LST in the title, especially if you decide to enhance the discussion about cause-and-effect between LST and albedo for the QEI.

Figure 1 caption – do you mean "Top-left inset" and "Bottom-right inset"? Figures 2, 3, 5, 6, 7 & 9 – it should be mentioned in the caption that white (Figs 2 & 3) is not ice and brown is not ice (Figs 5, 6, 7 & 9). Figure 8 – lower panel; I am confused about the point that you are trying to get across in this graph; please clarify

Line 14 – note that the range of years says 2008 – 2012 on line 486 60 – MODIS has already been defined on lines 53 & 54 76 & 78 – Lutch should be spelled Lucht 99 – perhaps the word "detectors" should be used instead of "instruments"?? 120 – 124 – this discussion of MOD10A1 should be deleted; it is not relevant to this paper, and MOD10A1 is never mentioned again in the paper; it really isn't good validation for

MCD43 151 – This is misleading because, in Hall et al. (2008a), the pixels when the temperature was >2 deg C were removed since it's impossible to have temperatures greater than zero and still be ice. Hall et al. (2008a) did not remove pixels with errors >2 deg C that were below zero, if I am recalling correctly. 155 – it should be stated that MOD11A2 Collection 6 data were downloaded 235 – I would say "opposite from" instead of "opposite to" 319 – I think Sections 3.3 and 4.1 could be shortened and made to be more concise 601 – should read IEEE instead of IEE

---

## Referee Comment (RC2) · Anonymous Referee #2 · 23 Sep 2017

TCD Review:

**Characterization of Canadian High Arctic glacier surface albedo from MODIS C6 data, 2001-2016**

C.A. Mortimer, M. Sharp

**General Comments:**

The manuscript presents an important effort in understanding Canadian High Arctic glacier albedo and surface temperature.  Change in Queen Elizabeth Islands (QEI) glacier surface albedo is presented from the combined Aqua and Terra MODIS Collection 6 albedo product and change in glacier surface temperature is presented from Terra MODIS Collection 5 Land Surface Temperature and Emissivity product.  The temporal period of the analyses include QEI summer months (June, July, August) from 2001-2016.   Albedo and land surface temperature changes are evaluated and presented as monthly and seasonal averages as well as spatial patterns.   Satellite analysis is compared with reanalysis data, some complementary observation data and studies published to date.

**Major Considerations:**

As written, the manuscript can be difficult to follow.  The Data and Methods section provides appropriate content and detail.  However, the manuscript may benefit from reorganization of content in the Introduction, Results, and Discussion sections.   Content in the Discussion section could be moved to the beginning of the manuscript, with the option to provide a 'Background' section.   Supporting details of complementary observations (e.g. snowfall, temperature) introduced in the Discussion might be better placed earlier in the manuscript, near results (see line by line suggestions below).  In addition, there are grammatical errors and smaller ordering inconsistencies throughout the manuscript that deter from readability.  Careful editing and attention to detail is needed in revising and ensuring the success of this manuscript.

In the data and methods section (Page 4, line 122) MOD10A1 data was mentioned "to be a reasonable representation of the true surface albedo".  Does this refer to analysis conducted by the authors for this manuscript or to Box et al., 2017?  Please clarify.  If MOD10A1 data was not inspected by the authors for this manuscript, why not?  Additionally, is there a reason the authors have focused on Terra data?  The manuscript may be strengthened by analysis of MODIS Aqua MYD10A1 and Terra MOD10A1 data.   As stated, the MCD43A3 data represents data acquired from both Terra and Aqua sensors.  Calibration challenges which have primarily impacted Terra data may be teased apart by analysis of Aqua MYD10A1 data, Terra MOD10A1 data, and combined sensor MCD43A3 data.

Section 2.2 discusses MODIS LST data.  Why was Collection 5 MOD11A2 data used? (Page 5, lines 155-157)  Communication with the MODIS data distributor, LP DAAC, revealed that MOD11A2 and MYD11A2 Collection 6 data have been available since mid-2015.  Additionally, with the discussion of the

Terra sensor degradation, it seems shortsighted to use Terra data only.  Why was Terra data used?  Was Aqua data used (i.e. MYD11A2)?  If not, why not?  Recommendation to add MYD11A2 data analysis.

Reorganization of content is advised toward readability.  There are several cases where content is difficult to follow.  Example, Results Section, 3.1   Suggestion to add sentences clarifying and further detailing results.  As written, parts of the section are terse and non-intuitive.  Lines 196-197 Why does Table 1 2001-2016 average differ from manuscript stated average (i.e. Table 1 states 0.599, manuscript states 0.550)?   For example, readers may appreciate a sentence stating that in addition to the JJA averages, monthly mean albedo was also calculated (Table 2).  Include references to Tables, methods as appropriate.

Similarly, in the Discussion section, it is stated that "delayed snowfall onset and limited melt in August were inferred from the GRACE mass change record".    However, earlier in the manuscript, speculative statements are made on snowfall patterns, implying there is no quantitative data nor analyses to site for QEI.  Use of a specific snowfall proxy data (i.e. GRACE, reanalysis) or other observational datasets would be more quantitative than speculative remarks currently in the Results and Discussions sections.  Two specific areas of speculation are: (1) Results Section 3.1, line 203, 'may indicate early onset of snowfall that fall' and (2) Results Section 3.2, lines 290-292 snowfall patterns may be changing in QEI).  Quantified data based statements are preferred over speculation where possible.

Recommendation for the authors to clearly state the additional QEI LST analysis provided in this manuscript as compared to Mortimer et al., 2016.  (See some line by line comments below.)  I recommend considering the role of the QEI LST analysis in this manuscript.

**Specific, Minor Comments:**

Was the Randolph Glacier Inventory used to delineate QEI glacier analysis areas?   If so, please state clearly where appropriate (i.e. manuscript text, figure captions) and cite.

Page 2, line 38 – What is meant by 'accelerated release'?  Suggestion to reword, clarify intent.

Page 2, lines 40-42 – The authors are correct to state fresh snow would raise surface albedo.  Suggestion to include QEI precipitation data quantifying the suggestion, or reference precipitation studies.

Page 2, line 49 – Suggestion to clarify for readers the source of the temperature analysis used in referenced studies.

Page 3, lines 89-91 – Suggestion to edit sentences.  As written, one could glean that it is difficult to discriminate between surface snow and ice vs cloud spectral visible-thermal infrared response.  This is not always difficult to do spectrally.  If the authors intend to discuss cloud remote sensing only, please clarify this.  A more appropriate reference than Hall 2008a, may be Hall et al., 2002, MODIS snow-cover products, Remote Sensing of Environment, 83, 181-194.

Page 4, line 98 – Delete 'some of'.

Page 4, line 102 – Lyapustin et al. was not the first to report on Terra's band degradation. Recommendation to additionally read and cite early / appropriate work, e.g. Xiong et al., 2001, Degradation of MODIS optics and its reflective solar bands calibration, doi: 10.1117/12.450646

Xiong and Barnes, 2006, An overview of MODIS radiometric calibration and characterization, doi: 10.1007/s00376-006-0008-3

Sun et al., 2014 Time-dependent response versus scan angle for MODIS reflective solar bands, IEEE TGRS, doi: 10.1109/TGRS.2013.2271448

Page 4, paragraph 3.  Suggestion for authors to reread literature on MODIS sensor calibration, degradation and capabilities.  Line 110, more correct to state that C6 did improve radiance measurements from launch to ~2013.  It remains to be assessed how accurate and reliable MODIS C6 data will be moving forward from C6 implementation (~2013 to present).  Recommendation to check the MODIS Characterization Support Team literature https://mcst.gsfc.nasa.gov/publications?f%5Btype%5D=102 .  For lines 113-114, it is not that the sensor is capable of identifying trends greater than 0.01, so much as +/- 0.01 is the limit of MODIS sensor accuracy and precision.  The paragraph could be rewritten to be more informative and clear regarding MODIS sensor design and capabilities.

Page 5, line 129 – Suggestion to move citation to correct location in the sentence, i.e. Schaaf and Wang 2015 reference should immediately follow MCD43 product mention.

Page 5, line 141 – Awkward as written, suggestion to reword to clarify further use of BSA term e.g. 'henceforth BSA refers to the black sky albedo MODIS shortwave broadband data.'

Page 5, lines 151-152 – The authors seem to generalize in that "uncertainties in the MOD11A2 LSTs arise mainly from cloud contamination".  Suggestion to reread relevant literature and present accurately. Does the sentence refer to over snow only?

Page 6, line 159 – Is there a reference from precipitation records/data in QEI? i.e. what station, record or data is 400 mm/yr derived from?

Page 7, line 195 – Suggestion to include MCD43A3 for clarity and completeness.

Page 9, line 272 – Note that it is not only the calibration accuracy that limits the capability to measure trends, but also the sensor design.   Please add reference to sensor capabilities (e.g. Justice et al., 1998, doi: 10.1109/36.701075 and/or similar on MODIS instrument design and post-launch capabilities, see https://mcst.gsfc.nasa.gov/publications?f%5Btype%5D=102 ).

Page 12, lines 365-366, Suggestion to reword sentence, avoiding use of "positive (negative)" words side by side.

Page 12, lines 376-377   It is stated that "delayed snowfall onset and limited melt in August were inferred from the GRACE mass change record".   Did the authors process GRACE data, or does this reference a study?   If it references a study, please include the citation.   If the authors processed GRACE data, please include mention of in Data and Methods.

Page 13, lines 407-409   Example of content that may be better placed in the beginning of the manuscript.

Page 15, lines 465-466   Example of sentence where it is difficult to assess what is new in this manuscript vs. Mortimer et al., 2016.

Figure 1 caption – Do the authors intend to reference Moderate Resolution Imaging Spectroradiometer instead of "Moderate Resolution Spectroradiometer"?

Figure 1 image – In some formats, it is difficult to differentiate the thematic colors used in the figure. Consider if there may be other colors to use.  For reader friendliness, it may also help to move the legend and increase the font size of the legend text.

Also, consider adding a label to the 8 regions.   This may help readers in interpretation of Supplemental Table S2.

Figure 4 – Figure text, misspelled word 'anomaly' in two locations, please correct.

Figure 5 – Why there is no data (white in figure) for the period 2001-2016 in some locations of QEI ice?

Figure S7 – Example of important, interesting QEI LST content for the authors to consider moving from the supplemental material to the main manuscript.

---

## Author Comment (AC1) · 14 Nov 2017

**Tc-2017-160. Author's response to RC2**

**Reviewer 2 (RC2):** Received and published 23 September 2017

**Reviewer 2's general comments:**
The manuscript presents an important effort in understanding Canadian High Arctic glacier albedo and surface temperature. Change in Queen Elizabeth Islands (QEI) glacier surface albedo is presented from the combined Aqua and Terra MODIS Collection 6 albedo product and change in glacier surface temperature is presented from Terra MODIS Collection 5 Land Surface Temperature and Emissivity product. The temporal period of the analyses includes the QEI summer months (June, July, August) from 2001-2016. Albedo and land surface temperature changes are evaluated and presented as monthly and seasonal averages as well as spatial patterns. Satellite analysis is compared with reanalysis data, some complementary observation data and studies published to date.

**AR**: We thank the reviewer for the thoughtful evaluation of our manuscript. We have reviewed and addressed both the major considerations and minor specific comments. This is outlined below.

- RC refers to reviewer's comment
- AR refers to author's response
- Revised sections of the manuscript are presented in *italics*.

**Reviewer 2's major considerations and author's responses**

**RC 1:** "As written, the manuscript can be difficult to follow. The Data and Methods section provides appropriate content and detail. However, the manuscript may benefit from reorganization of content in the Introduction, Results, and Discussion sections. Content in the Discussion section could be moved to the beginning of the manuscript, with the option to provide a 'Background' section. Supporting details of complementary observations (e.g. snowfall, temperature) introduced in the Discussion might be better placed earlier in the manuscript, near results (see line by line suggestions below). In addition, there are grammatical errors and smaller ordering inconsistencies throughout the manuscript that deter from readability. Careful editing and attention to detail is needed in revising and ensuring the success of this manuscript."

**AR 1:** The manuscript has been revised to make it more concise and easier to follow. Specifically, Sections 3.3 (now section 3.4) and 4.1 (now section 4) have been shortened. Repetition in sections 3.3 and 4.1 has been removed. Sections 4.1 and 4.2 have been merged (now section 4). Additional attention has been paid to grammar and ordering inconsistencies throughout the manuscript to improve readability. While we appreciate the suggestion of a background section we chose not to include one here. We felt that, after revising and condensing the manuscript, the addition of a background section was not necessary (see **RC/AR 29**).

**Revised text:**

**Revised Section 3.4, Lines 365-392:**

[revised manuscript text omitted]

**RC 2:** "In the data and methods section (Page 4, line 122) MOD10A1 data was mentioned "to be a reasonable representation of the true surface albedo". Does this refer to analysis conducted by the authors for this manuscript or to Box et al., 2017? Please clarify. If MOD10A1 data was not inspected by the authors for this manuscript, why not?"

**AC 2:** The reference to the MOD10A1 data has been removed and the methods section (section 2.1 MODIS sensor degradation) detailing MODIS sensor calibration issues has been revised.

**Revised text, lines 90-138:**

[revised manuscript text omitted]

**RC 3:** "Additionally, is there a reason the authors have focused on Terra data? The manuscript may be strengthened by analysis of MODIS Aqua MYD10A1 and Terra MOD10A1 data. As stated, the MCD43A3 data represents data acquired from both Terra and Aqua sensors. Calibration challenges which have primarily impacted Terra data may be teased apart by analysis of Aqua MYD10A1 data, Terra MOD10A1 data, and combined sensor MCD43A3 data."

**AR 3:** The combined MCD43A3 data product was selected for this study (instead of the Terra or Aqua only) to ensure optimal spatial coverage of the QEI ice cover. The study is meant as a first assessment of QEI surface albedo and is just as concerned with the spatial patterns in albedo just as it is with the temporal patterns

of albedo change. We acknowledge that these data include observations from both the Aqua and Terra sensors, and are therefore affected by calibration challenges that impact the Terra data more than the Aqua data. The methods section (Section 2.1 MODIS sensor degradation) has been re-written to provide a better account of the issues relating to sensor degradation.

**Revised text, lines 90-138:**

**'2.1.1 MODIS sensor degradation**

*The MODIS sensors are currently operating well beyond their expected [productive] six year lifetimes (Barnes et al, 1998; Justice 1998) and the detectors are degrading (Xiong et al., 2001). For both the MODIS Terra and Aqua sensors, instruments were calibrated pre-launch (radiometric, spatial, specular calibration) (Gunther et al., 1996). On-orbit calibration procedures were included to monitor the sensor degradation that is expected as the instruments are exposed to solar radiation (Gunther et al., 1996). For the reflective solar bands (0.41–2.2 µm) the on-board calibration system includes a solar diffuser (SD) calibrated using the solar diffuser stability monitor (SDSM) (Gunther et al., 1998). Lunar and Earth view observations (for select desert sites) are also used to assess radiometric stability (Sun et al., 2003). Even so, long-term scan mirror and wavelength dependent degradation, which are not sufficiently accounted for by the on-board calibrators (SD/SDMS), have been observed (Xiong et al., 2001; Lyapustin et al., 2014 and references therein). Calibration degradation effects, which are largely confined to the MODIS Terra sensor, are greatest in the blue band (B3) and decrease with increasing wavelength (Xiong and Barnes, 2006). An anomaly in the SD door operation (3 May 2003) and a decision to leave the door permanently open, exposing the SD to additional solar radiation, resulted in degradation of the SD on the MODIS Terra sensor that was faster than had originally been anticipated, and than was observed for the MODIS Aqua sensor (Xiong et al., 2005). Differences in the response versus scan angle (RVS) for the two side mirrors were also characterized pre-launch (Barnes et al., 1998). The RVS is important because it describes the scan mirror's response to different angles of incidence (AOI, for each band, detector and mirror side) (Sun et al., 2014). However, for the MODIS Terra sensor, following an overheating incident during pre-launch calibration, the RVS was not re-characterized and the exact pre-launch RVS characteristics are not known (Pan et al., 2007; Sun et al., 2014 and references therein). These issues have resulted in the performance of the MODIS Terra sensor being poorer than that of the MODIS Aqua sensor.*

*As a normal part of the operational procedure, the MODIS Characterization Support Team (http:mcst.gsfc.nasa.gov) periodically updates the calibration algorithms and approaches, during which time the entire L1B record (calibrated top of the atmosphere radiances) is re-processed to reflect improved understanding and characterization of changes to the instruments. Even so, non-physical trends in MODIS Terra data products, that result from calibration drift, have been observed and are well documented (e.g. Xiong et al., 2001; Xiong and Barnes, 2006; Franz et al., 2008; Kwiatkowski et al., 2008; Wang et al., 2012; Lyapustin et al., 2014; and references therein). The latest revision occurred with the C6 data and includes on-orbit calibration procedures to mitigate long-term calibration drift, particularly at the shorter wavelengths (Wenny et al., 2010; Toller et al., 2013; Sun et al., 2014; Lyapustin et al., 2014). The C6 dataset uses the on-board calibrators (e.g. SD/SDMS) and the mirror side ratios from lunar standard and Earth view observations (Toller et al, 2013; Sun et al., 2014). The C6 revision also includes an additional approach, aimed primarily at the short-wavelength bands, that uses observations of desert sites (pseudo-invariant targets) to derive instrument calibration coefficients and RVS at multiple AOIs (instead of the two AOIs provided by the SD and lunar standard) (Toller et al, 2013; Sun et al., 2014). Although this vicarious approach is less accurate than the one that uses the mirror-side ratios calibrated using a lunar standard, it has been found to provide a significant improvement to the L1B radiance measurements relative to the C5 data, prior to ~2013 (Toller 2013; Lyapustin et al., 2014). Updated L1B C6 radiances can be up to several percent higher than the C5 values (e.g. Band 3 and for most recent period ~2013 onward) (Toller et al., 2013; Lyapustin et al. 2014; Casey et al., 2017). However, evaluation of the L1B C6 Band 3 (0.46–0.48 µm) radiance over a desert site (Libya 4) identified residual errors (decadal trends on the order of several tenths of 1%; Lyapustin et al., 2014) that are within the product's stated accuracy (2% in absolute reflectance units for the reflective solar bands (Barnes et al., 1998; Justice et al., 1998)). The impact of the C6 updates on higher level MODIS*

*science products is difficult to quantify because the corrections are time, mirror-side, angular, and detector dependent (Toller et al., 2013; Lyapustin et al. 2014; Sun et al., 2014). In addition, the C6 revision includes updates to algorithms (in addition to the calibration updates) used in the derivation of specific higher-level products (https://www.umb.edu/spectralmass/terra_aqua_modis/v006 outlines changes made to the MCD43A3 C6 data product). Important for the current study, however, is that recent analysis of surface albedo over the Greenland Ice Sheet, immediately to the east of the QEI, using MODIS C6 data (including the MCD43A3 product used in this study) identified statistically significant albedo declines over the wet snow zone (Casey et al., 2017). For the most part, these declines are thought to be physically real (Casey et al., 2017), which gives us confidence in the albedo trends presented here. There are no long-term, spatially distributed, in situ albedo records from the glaciers and ice caps in the QEI, so a comparison between the MCD43A3 records and ground observations is not possible. This is both a motivation for and a limitation of, the current study.'*

**RC 4:** "Section 2.2 discusses MODIS LST data. Why was Collection 5 MOD11A2 data used? (Page 5, lines 155-157) Communication with the MODIS data distributor, LP DAAC, revealed that MOD11A2 and MYD11A2 Collection 6 data have been available since mid-2015. Additionally, with the discussion of the Terra sensor degradation, it seems short-sighted to use Terra data only. Why was Terra data used? Was Aqua data used (i.e. MYD11A2)? If not, why not? Recommendation to add MYD11A2 data analysis."

**AR 4:** The manuscript has been revised and now uses the C6 MOD11A2 data instead of the C5 product. Terra data were used because of the time-period under investigation (2001-16). Terra data are available since 2000 while Aqua data are only available after 2002. Use of the Terra LST data (MOD11A2) allowed comparison with the albedo record for the full 2001-16 time period.

**RC 5:** "Reorganization of content is advised toward readability. There are several cases where content is difficult to follow. Example, Results Section, 3.1 Suggestion to add sentences clarifying and further detailing results. As written, parts of the section are terse and non-intuitive."

**AR 5:** Section 3.1 has been re-ordered and additional text has been added for readability.

**Revised text, Section 3.1 (now lines 224-242):**

*'3.1 Mean summer albedo*
*Annual maps of the mean summer clear-sky broadband shortwave black-sky MCD43A3 albedo for all glacier-covered surfaces in the QEI for the 2001–16 period are presented in Figure 2. The QEI-wide mean summer BSA, averaged across all 16 years, was 0.550 ± 0.115 (mean ± 1 standard deviation; Table 1). The lowest QEI-wide mean summer BSA (0.539 ± 0.127) was recorded in 2011 while the highest (0.668 ± 0.089) was recorded in 2013 (Table 1).*
*In general, mean summer BSA is lower around the margins of the ice masses, where glacier ice is exposed in the summer, than it is in the higher elevation interior regions where snow or firn are exposed year-round (Fig. 2). During years when the QEI-wide mean summer BSA was low (e.g. 2011), we observed a broad zone of low albedo values (< 0.4) around the margins of the major ice masses (Fig. 2). Conversely, in years when the mean summer BSA was high (e.g. 2013), this zone was much less obvious. High data dropout at high elevations on Axel Heiberg Island, and over the summit of the Devon Ice Cap in 2014 and 2006 (Table S1), may have produced a negative albedo bias for these regions, since the albedo is typically greater at higher elevations. Aggregating the 2001–16 average mean summer BSA into 50 m elevation bins, we observed a linear rate of BSA increase with elevation (0.0085 per 50 m elevation bin, $r^2$ = 0.99).*
*In addition to the mean summer (JJA) BSA, the monthly mean BSA values for June, July, and August, were also investigated (Sect. 2.3) (Table 2, Fig. S1–S3). July had the lowest 16-year monthly mean BSA (0.551 ± 0.131), followed closely by August (0.579 ± 0.127) (Table 2). In each year during the 2001–16 period, the highest summer monthly BSA was always recorded in June while the lowest monthly BSA was recorded in either July or August. The lowest monthly mean BSA values for June and August were recorded in 2011, while*

*the lowest July mean BSA occurred in 2012. The highest monthly mean BSA for both June and July occurred in 2013; for August, it occurred in 2003.'*

**RC 6:** Lines 196-197 Why does Table 1 2001-2016 average differ from manuscript stated average (i.e. Table 1 states 0.599, manuscript states 0.550)?

**AR 6:** The value of 0.599 in Table 1 refers to the mean summer (JJA) BSA. The value of 0.551 on line 196-107 refers to the July mean BSA which is 0.551 ± 0.0131 and is presented in Table 2. As such, line 196-197 (**now lines 238-239**): *'July had the lowest 16-year monthly mean BSA (0.551 ± 0.131), followed closely by August (0.579 ± 0.127) (Table 2).'* remains unchanged except for addition of reference to appropriate Table (Table 2).

**RC 7:** For example, readers may appreciate a sentence stating that in addition to the JJA averages, monthly mean albedo was also calculated (Table 2, see Section 2.3).

**AR 7:** Section 3.1 has been re-ordered to make it easier to follow and a sentence (Line 209, highlighted) regarding the monthly mean BSA albedo has been added, with reference to the appropriate Table, Figures, and methods section.

**Revised text, section 3.1** (**now lines 224-242**): Line 209 highlighted

> *'3.1 Mean summer albedo*
> *Annual maps of the mean summer clear-sky broadband shortwave black-sky MCD43A3 albedo for all glacier-covered surfaces in the QEI for the 2001–16 period are presented in Figure 2. The QEI-wide mean summer BSA, averaged across all 16 years, was 0.550 ± 0.115 (mean ± 1 standard deviation; Table 1). The lowest QEI-wide mean summer BSA (0.539 ± 0.127) was recorded in 2011 while the highest (0.668 ± 0.089) was recorded in 2013 (Table 1).*
> *In general, mean summer BSA is lower around the margins of the ice masses, where glacier ice is exposed in the summer, than it is in the higher elevation interior regions where snow or firn are exposed year-round (Fig. 2). During years when the QEI-wide mean summer BSA was low (e.g. 2011), we observed a broad zone of low albedo values (< 0.4) around the margins of the major ice masses (Fig. 2). Conversely, in years when the mean summer BSA was high (e.g. 2013), this zone was much less obvious. High data dropout at high elevations on Axel Heiberg Island, and over the summit of the Devon Ice Cap in 2014 and 2006 (Table S1), may have produced a negative albedo bias for these regions, since the albedo is typically greater at higher elevations. Aggregating the 2001–16 average mean summer BSA into 50 m elevation bins, we observed a linear rate of BSA increase with elevation (0.0085 per 50 m elevation bin, $r^2 = 0.99$).*
> *==In addition to the mean summer (JJA) BSA, the monthly mean BSA values for June, July, and August, were also investigated (Sect. 2.3) (Table 2, Fig. S1–S3).== July had the lowest 16-year monthly mean BSA (0.551 ± 0.131), followed closely by August (0.579 ± 0.127) (Table 2). In each year during the 2001–16 period, the highest summer monthly BSA was always recorded in June while the lowest monthly BSA was recorded in either July or August. The lowest monthly mean BSA values for June and August were recorded in 2011, while the lowest July mean BSA occurred in 2012. The highest monthly mean BSA for both June and July occurred in 2013; for August, it occurred in 2003.'*

**RC 8:** Include references to Tables, methods as appropriate.

**AR 8:** Additional references to Tables, Figures, and methods have been added throughout These are identified on the markup copy, some examples are provided below.

**Examples:**

> **Results Section 3.1, lines 225-239:**

*'Annual maps of the mean summer clear-sky broadband shortwave black-sky MCD43A3 albedo for all glacier-covered surfaces in the QEI for the 2001–16 period are presented in Figure 2. The QEI-wide mean summer BSA, averaged across all 16 years, was 0.550 ± 0.115 (mean ± 1 standard deviation; Table 1). The lowest QEI-wide mean summer BSA (0.539 ± 0.127) was recorded in 2011 while the highest (0.668 ± 0.089) was recorded in 2013 (Table 1).*

*In general, mean summer BSA is lower around the margins of the ice masses, where glacier ice is exposed in the summer, than it is in the higher elevation interior regions where snow or firn are exposed year-round (Fig. 2). During years when the QEI-wide mean summer BSA was low (e.g. 2011), we observed a broad zone of low albedo values (< 0.4) around the margins of the major ice masses (Fig. 2). Conversely, in years when the mean summer BSA was high (e.g. 2013), this zone was much less obvious. High data dropout at high elevations on Axel Heiberg Island, and over the summit of the Devon Ice Cap in 2014 and 2006 (Table S1), may have produced a negative albedo bias for these regions, since the albedo is typically greater at higher elevations. Aggregating the 2001–16 average mean summer BSA into 50 m elevation bins, we observed a linear rate of BSA increase with elevation (0.0085 per 50 m elevation bin, $r^2 = 0.99$).*

*In addition to the mean summer (JJA) BSA, the monthly mean BSA values for June, July, and August, were also investigated (Sect. 2.3) (Table 2, Fig. S1–S3). July had the lowest 16-year monthly mean BSA (0.551 ± 0.131), followed closely by August (0.579 ± 0.127) (Table 2).'*

**Results Section 3.2.1, lines 245-250:**

*'The mean summer BSA anomalies, relative to the 2001–16 mean, are presented in Figure 3 and Table 1. For consistency with the regression analysis (Sect. 3.3), BSA anomalies were only computed for pixels having mean summer BSA observations in 11 or more of the 16 years (Sect. 2.3). The period 2001–16 is characterised by a six-year period of positive BSA anomalies (2001–06) followed by a six-year period of negative BSA anomalies (2007–12) (Table 1).'*

**Results Section 3.2.2, lines 285-287:**

*'In addition to the mean summer (JJA) BSA anomalies, the mean June, July, and August BSA anomalies were also examined (Fig. S4–S6) (Sect. 2.3). The sign of the mean July BSA anomaly (Table 2) was always consistent with that of the mean summer (JJA) BSA anomaly (Table 1).'*

**RC 9:** Similarly, in the Discussion section, it is stated that "delayed snowfall onset and limited melt in August were inferred from the GRACE mass change record". However, earlier in the manuscript, speculative statements are made on snowfall patterns, implying there are no quantitative data nor analyses to cite for QEI. Use of a specific snowfall proxy data (i.e. GRACE, reanalysis) or other observational datasets would be more quantitative than speculative remarks currently in the Results and Discussions sections.

**AR 9:** While we agree that quantified data based statements are preferred over speculation where possible, in the case of the QEI ice cover, spatially-distributed in-situ precipitation measurements do not exist for this time-period. Where appropriate, speculative comments have been removed from the text (**see RC/AR 28**). A reference to recent modeled precipitation changes (Vincent et al. 2015) (**RC/AR 14**) was added to the introduction. Vincent et al. (2015) found an increase in precipitation and an increase in the ratio of snowfall to total precipitation over the Canadian Arctic during the period 1948-2012. However, it is important to note that these trends were calculated by gridding climate station data (50km x 50km grid). There are only three such stations (Resolute Bay, Eureka, Alert) in the QEI and none of these are located over ice. Because of the coarse spatial resolution of the modelled data and the caveat that these trends were not validated over the permanent ice cover, the modelled data were not useful for validating trends at the scale of the individual ice caps which are the focus of our study.

We did consider using other modeled climate data in our study. The CanRCM4 (0.44°; 0.22° up to 2009) and MARv3.5.2 20 km monthly data forced with ERA Interim were investigated. However, we found the

spatial resolution of these data to be too coarse to discern spatial patterns in precipitation, and MAR results only covered a part of the QEI. Similar issues of spatial resolution were encountered with the NCEP/NCAR Reanalysis R1 temperature and precipitation data (2.5°x2.5° grid). Additionally, the absence of in-situ measurements on ice means that we are unable to validate these datasets. For the temperature record, it was possible to use in-situ climate records from stations located on the ice (Agassiz and Devon Ice Caps) to validate the NCEP/NCAR Reanalysis R1 data and MODIS LST records (Sharp et al., 2011). Sharp et al. (2011) found upper-air (700hPa) air temperatures from NCEP/NCAR Reanalysis R1 data to be well correlated with on-ice air temperature records, available for the Devon (1997-2009) and Agassiz (1988-2009) Ice Caps. Both Sharp et al. 2011 (2000-06) and Mortimer et al. 2016 (2000-15) found the 700hPa air temperatures to be strongly correlated with the MODIS LST record. A similar comparison is not possible for precipitation due to the absence of in situ measurements on ice, and caution was taken not to over-interpret precipitation trends.

The GRACE data do not provide detailed spatial information about snowfall patterns. However, when combined with existing knowledge of the accumulation and ablation seasons in the region these data can provide information about the relative timing of the transition from summer mass loss (which ends when the minimum annual glacier mass is reached) to the winter period of mass gain (inferred from the onset of mass increase) which occurs in autumn. This is described in **lines 454-461** of the revised manuscript.

**Revised text, lines 454-461:**

> *'To investigate the relationship between the albedo record and variability in precipitation we examined the cumulative mass change record for the QEI from the Gravity Recovery and Climate Experiment (GRACE; Wolken et al. (2016), extended to 2016 (B. Wouters, personal communication, 2017)). The record shows that in 2006, 2010, and 2016 (the years when PC2 of the 16-year mean summer BSA record had the largest EOFs of any Principal Component, Sect. 3.4), once the annual minimum glacier mass was reached, there was a prolonged period of constant low mass (i.e. no melt or snowfall) before fall/winter accumulation began (inferred from an increase in mass). In other years, there was a sharp transition from the local end-of-summer mass minimum to the period of seasonally increasing mass.'*

**RC 10:** Two specific areas of speculation are: (1) Results Section 3.1, line 203, 'may indicate early onset of snowfall that fall' and (2) Results Section 3.2, lines 290-292 snowfall patterns may be changing in QEI). Quantified data based statements are preferred over speculation where possible.

**AC 10:** Again, while we agree that quantitative statements are preferred over speculation, the absence of in-situ precipitation data limits our ability to provide them. The specific areas of concern noted in RC 10 have been removed from the manuscript.

**RC 11:** Recommendation for the authors to clearly state the additional QEI LST analysis provided in this manuscript as compared to Mortimer et al., 2016. (See some line by line comments below.) I recommend considering the role of the QEI LST analysis in this manuscript.

**RC11a:** Recommendation for the authors to clearly state the additional QEI LST analysis provided in this manuscript as compared to Mortimer et al., 2016. (See some line by line comments below.)

**AR 11a:** Following your recommendations, the manuscript has been revised and now uses the MOD11A2 C6 data instead of the C5 data. Text has been added to the Methods (Section 2.2) to clearly state the differences in the QEI LST analysis provided in this manuscript compared to Mortimer et al., 2016.

**Revised additional text, lines 175-177:**

*'The LST analyses presented here are an update of those presented by Mortimer et al. (2016) which used MODIS C5 data. The two analyses also differ in the time period used (we use 2001–16 instead of 2000–15 to coincide with the BSA analysis).'*

**RC 11b:** I recommend considering the role of the QEI LST analysis in this manuscript.

**AR 11b:** Temperature changes drive the snow grain metamorphism and melt of snow which both affect its albedo. In addition, the removal of the seasonal snowpack (which exposes lower albedo firn or glacier ice), and the melt of glacier ice (which releases impurities which have low albedo and thus changes the surface albedo of both ice and snow) are driven by changing air and surface temperatures. Hence comparison of the LST and albedo datasets provides insight into the mechanisms driving albedo change in different regions of the QEI. As such, analyses of the LST dataset are included in this manuscript to help understand the observed spatiotemporal patterns of albedo change in the QEI. Additional text clarifying the reason for inclusion of the LST data has been added to both the methods (Section 2.2) and results (Section 3.5) sections, as well as to the conclusion.

**Revised text:**

**Methods Section 2.2 lines 161-167 added text:**

*'Warmer surface temperatures increase the rates of grain metamorphism and snowmelt, resulting in larger snow grains which have a lower albedo than those of fresh snow (Wiscombe and Warren 1980; Colbeck 1982) (Sect. 1). Air and surface temperatures also affect the timing of removal of the seasonal snowpack, which exposes lower albedo firn or glacier ice. Additionally, the melt of glacier ice releases impurities that have a low albedo and thus change the surface albedo of both ice and snow (e.g. Clarke and Noone 1985; Doherty et al., 2010; Sect. 1). As such, analysis of the glacier surface temperature and comparison of these data with the albedo record is included to help understand the observed spatiotemporal patterns of glacier albedo change in the QEI.'*

**Results section 3.5 lines 394-396 added text:**

*'Owing to the positive feedback between albedo and surface temperature we would expect to observe strong increases in surface temperature where albedo declines were large. To investigate the relationship between temperature and albedo over the QEI, the 16-year mean summer LST record was compared with the 16-year mean summer BSA record (Sect. 2.3).'*

**Conclusion, added text (lines 476-485):**

*'Albedo declines increase the proportion of incoming solar radiation absorbed at the air-ice interface, and thus the energy available to drive melt, warming, and further surface albedo decline. Warmer temperatures, in turn, increase the rate of snow grain metamorphism which lowers the albedo. Air and surface temperatures affect the removal (timing and extent) of the seasonal snowpack which exposes lower albedo firn and/or glacier ice, while melting glacier ice releases impurities that further reduce its albedo. Given that surface temperature and albedo are inextricably linked, knowing where and when albedo changes are likely to occur in future is important for predicting future rates of mass loss from the QEI ice caps. Recent investigations of atmospheric circulation patterns over the QEI (e.g. Gardner and Sharp, 2007) focused on characterization of July temperature and atmospheric conditions, since July is usually the month when melt rates peak. Our results suggest, however, that changes occurring during the month of August are also important, especially as the length of the melt season continues to increase.'*

**RC 12:** Was the Randolph Glacier Inventory used to delineate QEI glacier analysis areas? If so, please state clearly where appropriate (i.e. manuscript text, figure captions) and cite.

**AR 12:** The Randolph Glacier Inventory was used to delineate QEI glacier areas throughout the analysis. This was stated in **Line 186 (now line 220-222)** of the original manuscript: *'To ensure that only data for glaciated surfaces were retained, all BSA and LST outputs used in this analysis were clipped to the Randolph Glacier Inventory v3.2 region 32 (Arctic Canada North) reference polygons (Arendt et al., 2012; Pfeffer et al., 2014).'*

**RC 13:** Page 2, line 38 – What is meant by 'accelerated release'? Suggestion to reword, clarify intent.

**AR 13:** Sentence reworded.

**Revised text, Line 36-38 (now lines 37-40):**
> *'Albedo decreases can also be caused by aerosol deposition (Warren and Wiscombe, 1980), biological activity on glacier surfaces (Fountain et al., 2004), and the release of impurities contained within melting snow and ice, which become concentrated at the snow/ice surface (Clarke and Noone, 1985; Conway et al., 1996; Flanner et al., 2007; Doherty et al., 2010).'*

**RC 14:** Page 2, lines 40-42 – The authors are correct to state fresh snow would raise surface albedo. Suggestion to include QEI precipitation data quantifying the suggestion, or reference precipitation studies.

**AR 14:** Reference to Vincent et al. (2015), who reported both an increase in total precipitation and an increase in the ratio of snowfall to total precipitation over the Canadian Arctic during the period 1948-2012, has been added. These trends were calculated by gridding climate station data. In the QEI, there are only three such stations (Resolute Bay, Eureka, Alert) and none of them are located on glacier ice (see **RC/AR 9**). Reference to Box et al. (2012) who found a second-order negative albedo feedback (defined by a positive correlation between albedo and surface air temperature anomalies) in the accumulation area of the nearby Greenland Ice Sheet has also been added to the end of this sentence.

**Revised text:**
> **Lines 39-42 (now lines 40-43)**
> *'Given the observed increases in air and glacier surface temperatures across the QEI (Mortimer et al., 2016) we anticipate a reduction in the surface albedo in this region, unless warming has also been accompanied by an increase in solid precipitation (Vincent et al., 2015) that is large enough to raise the surface albedo (e.g. Box et al., 2012).'*
> **Reference added**
> *Vincent, L. A., Zhang, X., Brown, R. D., Feng, Y., Mekis, E., Milewska, E. J., Wan, H., and Wang, X. L.: Observed trends in Canada's climate and influence of low-frequency variability modes, Journal of Climate, 28, 4545-4560, doi: 10.1175/JCLI-D-14-00697.1, 2015.*

**RC 15:** Page 2, line 49 – Suggestion to clarify for readers the source of the temperature analysis used in referenced studies.

**AR 15: Text revised for clarity.**

**Revised line 49 (now lines 49-52):**
> *'Between 2000 and 2015, increases in QEI summer 700 hPa air temperatures derived from the NCEP/NCAR R1 Reanalysis (Kalnay and others, 1996) and glacier surface temperatures from the Moderate Resolution Imaging Spectroradiometer (MODIS) were greatest in the north and west of the QEI (Sharp et al., 2011; Mortimer et al., 2016). We do not know whether there is a similar spatial pattern in the albedo record.'*

**RC 16:** Page 3, lines 89-91 – Suggestion to edit sentences. As written, one could glean that it is difficult to discriminate between surface snow and ice vs cloud spectral visible-thermal infrared response. This is not

always difficult to do spectrally. If the authors intend to discuss cloud remote sensing only, please clarify this. A more appropriate reference than Hall 2008a, may be Hall et al., 2002, MODIS snow-cover products, Remote Sensing of Environment, 83, 181-194.

**AR 16:** Text moved and re-written for clarity. Reference to Hall et al. (2002) has been added. The issue of concern here is the presence of clouds which can potentially introduce additional variability in the dataset (due to the removal of data when clouds are detected) that is not representative of physical changes on the surface.

**Revised text (now lines 139-144):**
> *'Furthermore, the MCD43A3 data used here were produced only under clear sky conditions (Hall et al., 2002; 2012). A conservative cloud mask is applied to remove observations made when clouds are detected (Ackerman et al., 1998). The resulting data gaps may introduce variability in the albedo record that is not representative of true physical change. Despite this, the MCD43A3 albedo product has been found to provide a reasonable representation of the seasonal albedo cycle over glaciers and ice caps (e.g. Stroeve et al., 2006). Hence, in the absence of long-term ground measurements of glacier surface albedo in the QEI, we made the assumption that this is also the case in the QEI.'*

> **Reference added:** *Hall, D. K., Riggs, G. A., Salomonson V. V. DiGirolamo N. E., Bayr K. J.: MODIS snow-cover products, Remote Sens. Environ., 83(1), 181-194, doi: 10.1016/S0034-4257(02)00095-0, 2002.*

**RC 16:** Page 4, line 98 – Delete 'some of'.

**AR 16:** Deleted 'some of'

**Revised text Line 98 (now lines 91-92):**
> *'The MODIS sensors are currently operating well beyond their expected [productive] six year lifetimes (Barnes et al, 1998; Justice 1998) and the detectors are degrading (Xiong et al., 2001).'*

**RC 17:** Page 4, line 102 – Lyapustin et al. was not the first to report on Terra's band degradation. Recommendation to additionally read and cite early / appropriate work, e.g. Xiong et al., 2001, Degradation of MODIS optics and its reflective solar bands calibration, doi: 10.1117/12.450646
Xiong and Barnes, 2006, An overview of MODIS radiometric calibration and characterization, doi: 10.1007/s00376-006-0008-3
Sun et al., 2014 Time-dependent response versus scan angle for MODIS reflective solar bands, IEEE TGRS, doi: 10.1109/TGRS.2013.2271448

**AR 17:** References to earlier studies have been added following additional review of the relevant literature (see **RC/AR 18**). References to Sun et al. (2014) and Xiong and Barnes (2006) are included in a revised section on MODIS sensor degradation (**lines 90-138).**

**Revised text:**

> **Line 102 (now line 91-92):**
> *'The MODIS sensors are currently operating well beyond their expected [productive] six year lifetimes (Barnes et al, 1998; Justice 1998) and the detectors are degrading (Xiong et al., 2001).'*

> **Revised methods section 2.1 (lines 90-138) text with references to** Xiong et al. (2001), Xiong and Barnes (2006) and Sun et al. (2014) are highlighted:

[revised manuscript text omitted]

**RC 18:** Page 4, paragraph 3. Suggestion for authors to reread literature on MODIS sensor calibration, degradation and capabilities. Line 110, more correct to state that C6 did improve radiance measurements from launch to ~2013. It remains to be assessed how accurate and reliable MODIS C6 data will be moving

forward from C6 implementation (~2013 to present). Line 110: Recommendation to check the MODIS Characterization Support Team literature https://mcst.gsfc.nasa.gov/publications?f%5Btype%5D=102 .

**AR 18:** Literature has been reviewed and have been included in the revised methods section (Section 2.1.1) presented above. Specific mention of the improved radiances up until ~2013 has been added.

**Revised text, Line 110 (now lines 121-123):**
> *'Although this vicarious approach is less accurate than the one that uses the mirror-side ratios calibrated using a lunar standard, it has been found to provide a significant improvement to the L1B radiance measurements relative to the C5 data, prior to ~2013 (Toller 2013; Lyapustin et al., 2014).'*

**RC 19:** For lines 113-114, it is not that the sensor is capable of identifying trends greater than 0.01, so much as +/- 0.01 is the limit of MODIS sensor accuracy and precision. The paragraph could be rewritten to be more informative and clear regarding MODIS sensor design and capabilities.

**AC 19:** This paragraph has been re-written to be more informative and accurate regarding MODIS sensor design and capabilities. The text *'is capable of identifying trends greater than 0.01'* has been removed.

**Revised methods text is included with RC/AR 20, below.**

**RC 20:** The paragraph (Page 4, Paragraph 3) could be rewritten to be more informative and clear regarding MODIS sensor design and capabilities.

**AR 20:** MODIS Characterization Support Team literature has been reviewed and this section has been re-written to be more informative.

**Revised text, lines 90-138:**

[revised manuscript text omitted]

**RC 21:** Page 5, line 129 – Suggestion to move citation to correct location in the sentence, i.e. Schaaf and Wang 2015 reference should immediately follow MCD43 product mention.

**AR 21:** Citation moved to earlier in the sentence.

**Revised Line 129 (now lines 147-149):**

*'Summer (1-2 June (day 152) to 30-31 August (day 243)) MODIS MCD43A3 and MCD43A2 C6 (Schaaf and Wang, 2015) data for MODIS tiles h17v00, h16v00, h16v01, and h15v01 for the period 2001-2016 were obtained from the NASA/USGS Land Processes Distributed Active Archive Center (http://lpdaac.usgs.gov/accessed November 2016).'*

**RC 22:** Page 5, line 141 – Awkward as written, suggestion to reword to clarify further use of BSA term e.g. 'henceforth BSA refers to the black sky albedo MODIS shortwave broadband data.'

**AR 22:** Line no longer included in manuscript.

**RC 23:** Page 5, lines 151-152 – The authors seem to generalize in that "uncertainties in the MOD11A2 LSTs arise mainly from cloud contamination". Suggestion to reread relevant literature and present accurately. Does the sentence refer to over snow only?

**AR 23:** This sentence referred to uncertainties in the trends derived from the MOD11A2 data and not in the MOD11A2 data itself. The text has been revised for clarity.

**Revised text, lines 151-152 (now lines 179-183):**

*'Uncertainties in trends derived from the MOD11A2 LST data arise mainly from cloud contamination (Box et al., 2012; Hall et al., 2012) and the removal of observations for periods when clouds are detected (Ackerman et al., 1998; Hall et al., 2008b). Variability in the number of clear-sky days within each observation period and from one year to the next was not found to*

*introduce significant variability in the MODIS-derived LST relative to the true near-surface air temperature in the QEI (see Mortimer et al., 2016).'*

**RC 24:** Page 6, line 159 – Is there a reference from precipitation records/data in QEI? i.e. what station, record or data is 400 mm/yr derived from?

**AR 24:** The reference for the 400 mm/yr should be Braithwaite, 2005. In the original manuscript, this reference, which refers to both the temperature and precipitation ranges, was included at the end of the following sentence. The text has been revised to clarify this omission.

**Revised text line 159 (now lines 187-188):**

*'Annual precipitation in the QEI is low (<400 mm yr$^{-1}$) and varies little from one year to the next; in contrast, the annual temperature range is large (> 40°C) (Braithwaite, 2005).'*

**RC 25:** Page 7, line 195 – Suggestion to include MCD43A3 for clarity and completeness.

**AR 25:** Mention of MCD43A3 included for clarity and completeness, as suggested.

**Revised text line 195 (now line 225-226):**

*'Annual maps of the mean summer clear-sky broadband shortwave black-sky MCD43A3 albedo for all glacier-covered surfaces in the QEI for the 2001–16 period are presented in Figure 2.'*

**RC 26:** Page 9, line 272 – Note that it is not only the calibration accuracy that limits the capability to measure trends, but also the sensor design. Please add reference to sensor capabilities (e.g. Justice et al., 1998, doi: 10.1109/36.701075 and/or similar on MODIS instrument design and post-launch capabilities, see https://mcst.gsfc.nasa.gov/publications?f%5Btype%5D=102 ).

**AR 26**: Sentence revised for accuracy in both the methods (Section #, line 202-205) and results (Section 3.3.1, line 315) sections. Additional references to literature discussing sensor capabilities (Barnes et al., 1998; Justice et al., 1998) have been added. References to the appropriate methods section have been included in the results.

**Revised text:**

**Line 125-128 (Section 2.1.1):**

*'…evaluation of the L1B C6 Band 3 (0.46–0.48 µm) radiance over a desert site (Libya 4) identified residual errors (decadal trends on the order of several tenths of 1%; Lyapustin et al., 2014) that are within the product's stated accuracy (2% in absolute reflectance units for the reflective solar bands (Barnes et al., 1998; Justice et al., 1998)).*

**Lines 192-195 (now lines 202-205, Section 2.3):**
*'Following Casey et al. (2017), BSA trends between -0.001 yr$^{-1}$ and +0.001 yr$^{-1}$ are considered to be negligible. Negligible trends were defined by Casey et al. (2017) on the basis of the magnitude of the residual calibration uncertainties in the C6 data (on the order of several tenths of one percent in TOA reflectance (Lyaputsin et al., 2014)) over pseudo-invariant desert sites (Sect. 2.1.1).'*

**Lines 298-299 (now Results Section 3.3.1 line 315):**
*'More than 95% of pixels experienced a non-negligible (> |0.001| yr$^{-1}$, Sect. 2.3) decrease in summer BSA…'*

**References added:**
*Barnes, W. L., Pagano, T.S., and Salomonson V.V.: Prelaunch characteristics of the Moderate Resolution Imaging Spectroradiometer (MODIS) on EOS-AM1, IEEE Transactions on Geoscience and Remote Sensing, 36(4), 1088-1100, doi: 10.1109/36.700993, 1998.*

*Justice, C.O., Vermote, E., Townshend, J.R.G., Defries, R., Roy, D. P., Hall, D. K., Vincent V. Salomonson V. V. et al.: The Moderate Resolution Imaging Spectroradiometer (MODIS): Land remote sensing for global change research, IEEE Transactions on Geoscience and Remote Sensing, 36(4), 1228-1249, doi: 10.1109/36.701075, 1998.*

**RC 27:** Page 12, lines 365-366, Suggestion to reword sentence, avoiding use of "positive (negative)" words side by side.

**AR 27:** This line, as written, is no longer included in the revised manuscript.

**RC 28:** Page 12, lines 376-377 It is stated that "delayed snowfall onset and limited melt in August were inferred from the GRACE mass change record". Did the authors process GRACE data, or does this reference a study? If it references a study, please include the citation. If the authors processed GRACE data, please include mention of in Data and Methods.

**AR 28:** Text has been re-organized for clarity. The above-mentioned delayed snowfall onset and limited melt in August were inferred from the GRACE mass change record data presented in Wolken et al. (2016), extended to 2016. The explanation of this was originally included in Section 3.3 (now section 3.4). The more detailed explanation has now been moved to the discussion and has replaced the general comment to better describe how we used the GRACE data.

**Revised text (lines 454-462):**
> *'To investigate the relationship between the albedo record and variability in precipitation we examined the cumulative mass change record for the QEI from the Gravity Recovery and Climate Experiment (GRACE; Wolken et al. (2016), extended to 2016 (B. Wouters, personal communication, 2017)). The record shows that in 2006, 2010, and 2016 (the years when PC2 of the 16-year mean summer BSA record had the largest EOFs of any Principal Component, Sect. 3.4), once the annual minimum glacier mass was reached, there was a prolonged period of constant low mass (i.e. no melt or snowfall) before fall/winter accumulation began (inferred from an increase in mass). In other years, there was a sharp transition from the local end-of-summer mass minimum to the period of seasonally increasing mass. This could indicate that in some years during the 2001–16 period, variability in August snowfall may have influenced the mean summer BSA.'*

**RC 29:** Page 13, lines 407-409 Example of content that may be better placed in the beginning of the manuscript.

**AR 29:** We appreciate your recommendation to include a background section or to include some of this material at the beginning of the manuscript. In revising the manuscript, however, we found that this particular point fitted better towards the end as it helps to place the importance and relevance of the August BSA change in the context of existing literature.

**RC 30:** Page 15, lines 465-466 Example of sentence where it is difficult to assess what is new in this manuscript vs. Mortimer et al., 2016.

**AR 30:** The text from Lines 465-466 is no longer included in the revised manuscript. However, in response to this comment we reviewed the existing text to ensure that it was clear as to whether we were referring to the current study or to Mortimer et al. 2016. Specifically, in response to your comment regarding C5 vs C6 (**RC 4**), LST anomalies were recalculated and are presented in Figure S7. Additional text has been included to specify differences between Mortimer et al. 2016 and this study. Where possible, references to figures in the current text (i.e. Figure 7 and Figure S7) are used instead of referring to the earlier Mortimer et al. 2016 study.

**Revised text and figures:**

**Added text lines 176-179:**

*'The LST analyses presented here are an update of those presented by Mortimer et al. (2016) which used MODIS C5 data. The two analyses also differ in the time-period used (we use 2001-16 instead of 2000-15 to coincide with the BSA analysis).'*

**Figure S7 (added Figure):**

[Figure]

**Figure S7: Mean August clear-sky BSA anomaly relative to the 2001–16 mean for the QEI ice caps relative to the 2001–16 mean. White areas outside of the ice caps indicate non-glaciated ice cover.**

Where possible, references to figures in the current text (i.e. Figure 7 and Figure S7) are used instead of referring to the earlier Mortimer et al. 2016 study.

> **Revised line 273-276:**
> *'In 2005 and 2006 (years when the mean summer QEI-wide BSA anomaly was near zero and there was a large amount of spatial variability in the mean summer BSA), BSA anomalies on the western-most part of Axel Heiberg Island (Fig. 3) were double the magnitude of the QEI-wide BSA anomaly (Table 1) and of the same sign. A similar feature (large anomaly values of the same sign as the QEI-wide anomaly) was also observed in the LST anomaly record in 2005 (Fig. S7).'*
> **Revised line 339-340:**
> *'…even though strong increases in LST (>0.1°C yr$^{-1}$) were observed in these locations between 2001 and 2016 (Fig. 7).'*

**RC 31:** Figure 1 caption – Do the authors intend to reference Moderate Resolution Imaging Spectroradiometer instead of "Moderate Resolution Spectroradiometer"?

**AR 31:** Figure 1 Correct, it should read Moderate Resolution Imaging Spectroradiometer instead of Moderate Resolution Spectroradiometer. In revised manuscript, MODIS is used in the figure caption instead of used instead of Moderate Resolution Imaging Spectroradiometer.

**Revised Figure 1 caption:**
> *'Figure 1: Glaciated regions of the Queen Elizabeth Islands. Background image: MODIS 4 July 2011. Inset: red polygon shows location of Queen Elizabeth Islands, Arctic Canada.'*

**RC 32:** Figure 1 image – In some formats, it is difficult to differentiate the thematic colours used in the figure. Consider if there may be other colours to use. For reader friendliness, it may also help to move the legend and increase the font size of the legend text. Also, consider adding a label to the 8 regions. This may help readers in interpretation of Supplemental Table S2

**AR 32:** Figure 1 revised. Bottom-right inset no longer included because regional BSA values are no longer referred to in text. Labels for the major ice masses have been included on the main map. Supplemental Table 2 has been removed.

**Revised Figure 1:**

[Figure]

**Figure 1: Glaciated regions of the Queen Elizabeth Islands. Background image: MODIS 4 July 2011. Inset: red polygon shows location of Queen Elizabeth Islands, Arctic Canada.**

**RC 33:** Figure 4 – Figure text, misspelled word 'anomaly' in two locations, please correct.

**AR 33:** Figure 4 corrected for two erroneous spellings of 'anomaly'

**Revised Figure 4:**

[Figure]

**Figure 4: Mean summer (JJA) composite NCEP/NCAR Reanalysis 500 hPa geopotential height anomaly for (a) a period of large negative BSA anomalies (2001–04) and (b) a period of large positive BSA anomalies (2007–12). Source: https://www.esrl.noaa.gov/psd.**

**RC 34:** Figure 5 – Why there is no data (white in figure) for the period 2001-2016 in some locations of QEI ice?

**AR 34:** Cloud cover results in an absence of data. The amount of missing data, presented in Table S1, varies between months and from one year to the next. Because of this variability, thresholds were set to ensure consistency between years in calculating the mean summer and mean monthly BSA. This was outlined in Section 2.3 **lines 191-194**: *'For each year during the 2001–16 period, mean summer (JJA) BSA was calculated for pixels having at least 10 BSA observations in each month (June, July, August) and at least 45 of a possible 92 observations during the JJA period. These monthly thresholds ensure both an even distribution of BSA data throughout the summer season and consistency between different years.'*

Further, to ensure the trend in BSA was not biased towards extremely warm or cold years, regressions were only computed for pixels having at least 11 mean summer BSA observations. This was outlined in **lines 197-202:** *'The mean summer (JJA) BSA and LST and the mean monthly (June, July, August) BSA, as well as the BSA and LST anomalies, were calculated on a pixel-by-pixel basis relative to the 2001–16 mean for pixels having mean summer observations in 11 or more years. These constituted ~87% and ~98% of possible BSA and LST pixels, respectively. Long-term rates of change in BSA and LST over the period 2001–16 were determined by linear regression between the 16 year records of mean summer LST/BSA and time. Consistent with the BSA and LST anomalies, regressions were computed on a pixel-by-pixel basis for all pixels having mean summer observations for 11 or more years.'*

**RC 35:** Figure S7 – Example of important, interesting QEI LST content for the authors to consider moving from the supplemental material to the main manuscript.

**AR 35:** Figure S7, which was revised to use the C6 instead of the C5 data (see **RC/AR 3**), has been moved from supplemental material to the main text and is now Figure 7.

**Figure 7 (revised and moved):**

[Figure]

Figure 7: QEI (a) 16 year average mean summer land surface temperature (°C) and (b) linear rate of change of mean summer LST (°C yr$^{-1}$) for the period 2001-16 over the QEI ice caps. Background image: MODIS, 4 July 2011. Brown indicates non-glaciated land cover.

---

## Author Comment (AC2) · 14 Nov 2017

**Tc-2017-160**

**Response to RC1**

**Reviewer 1 (RC1),** Received and published 13 September 2017.

Reviewer 1's general comments:

In this paper, the albedo and surface temperature of the glaciers in the Queen Elizabeth Islands (QEI) are investigated using MODIS data products for 2001 – 2016. Over the study period BSA is shown to decrease significantly in the month of July, while LST increases during the summer months (JJA). This is a valuable paper and it is well written (though a bit long) and technically sound. My comments below are almost all optional.

I believe that the authors have not fully taken advantage of this opportunity to explore the relationship between albedo and LST. Though they point out that albedo and LST are negatively correlated for the QEI, they do not discuss that relationship as cause-and-effect in either the Abstract or the Conclusion. The relationship between albedo and LST is discussed in Section 4.2 but I would have liked that discussion to have been better integrated throughout the paper. Since there was quite a bit of information about LST in the paper, the authors might want to consider including LST in the title of the manuscript.

The authors say that only the month of July shows statistically-significant albedo decrease over the study period. Thus, they should consider stating that the July albedo is decreasing over the study period instead of saying mean summer (JJA) albedo is decreasing.

If the paper can be made more concise it would be easier to follow.

**Author's response to R1 general comment:**

We thank the reviewer for the thoughtful evaluation of our manuscript. We have reviewed and addressed both the major considerations and minor specific comments.

1. **RC** refers to reviewer's comment
2. **AR** refers to author's response
3. Revised sections of the manuscript are presented in *italics*.

**RC**: If the paper can be made more concise it would be easier to follow.

**AR**: The paper has been revised to make it more concise and easier to follow. Sections 3.3 (now section 3.4) and 4.1 (now section 4) have been shortened. Repetition in sections 3.3 and 4.1 has been removed. Sections 4.1 and 4.2 have been merged (now section 4).

**Revised text:**

**Revised Section 3.4, Lines 365-392:**

[revised manuscript text omitted]

**RC**: Though they point out that albedo and LST are negatively correlated for the QEI, they do not discuss that relationship as cause-and-effect in either the Abstract or the Conclusion. The relationship between albedo and LST is discussed in Section 4.2 but I would have liked that discussion to have been better integrated throughout the paper.

**AR**: Following a comment from an anonymous reviewer the LST analysis was revised to use MODIS C6 data instead of MODIS C5 data. The MOD11A2 C6 product has a larger amount of missing data than the C5 product. This resulted in a larger number of no-data pixels in our BSA/LST correlations (new: 46% versus old 80%) which limited our ability to confidently interpret some of the spatial patterns that were discussed in our initial submission. As such, the discussion of the relationship between albedo and temperature has been reduced compared to the original manuscript and the discussion of cause-and-effect is also reduced. However, we have tried to better integrate discussion of LST and albedo earlier in the manuscript.

Additional linkages between albedo and LST have been added earlier in the manuscript. For example, Section 3.2 has now been divided into two separate sub-sections and linkages between albedo, LST, and glacier mass change (highlighted text) have been added to Section 3.2.1. Discussion of the links between

albedo and temperature have also been added to the methods section (Sect. 2.2) where we describe the MODIS LST data and the reason for its inclusion in the manuscript, and in the results Section 3.5 (comparison of albedo and LST patterns and trends). For brevity, only a short mention of the effect that the negative ice-albedo feedback could have on the QEI ice mass is included in the Abstract. Due to the change in data set used (MOD11A2 C6) and the resulting increase in the number of cells without data, the discussion of cause and effect is reduced. Additional text regarding the relationship between temperature and albedo has been included in the conclusion.

**Revised text**

**Section 2.2 MODIS LST (MOD11A2) added text (lines 161-168):**

> *'Warmer surface temperatures increase the rates of grain metamorphism and snowmelt, resulting in larger snow grains which have a lower albedo than those of fresh snow (Wiscombe and Warren 1980; Colbeck 1982) (Sect. 1). Air and surface temperatures also affect the timing of removal of the seasonal snowpack, which exposes lower albedo firn or glacier ice. Additionally, the melt of glacier ice releases impurities that have a low albedo and thus change the surface albedo of both ice and snow (e.g. Clarke and Noone 1985; Doherty et al., 2010; Sect. 1). As such, analysis of the glacier surface temperature and comparison of these data with the albedo record is included to help understand the observed spatiotemporal patterns of glacier albedo change in the QEI.'*

**Section 3.2.1 (lines 244-256), highlighted text moved from later in the manuscript and text edited for clarity:**

> *'Section 3.2.1 Mean summer (JJA) BSA anomalies*
>
> *The mean summer BSA anomalies, relative to the 2001–16 mean, are presented in Figure 3 and Table 1. For consistency with the regression analysis (Sect. 3.3), BSA anomalies were only computed for pixels having mean summer BSA observations in 11 or more of the 16 years (Sect. 2.3). The period 2001–16 is characterised by a six-year period of positive BSA anomalies (2001–06) followed by a six-year period of negative BSA anomalies (2007–12) (Table 1). Positive BSA anomalies were also observed in 2013 (+0.060) and 2014 (+0.015), while 2015 (-0.022) and 2016 (-0.005) saw a return to negative anomalies (Table 1). Negative BSA anomalies during the period 2007–12, which indicate a larger absorbed fraction of incoming shortwave radiation relative to the 16-year mean, coincide, and are consistent with, positive summer air and glacier surface temperature anomalies from 2007 to 2012 (Mortimer et al., 2016). Higher temperatures increase the rate of snow grain metamorphism which lowers the surface albedo (Wiscombe and Warren, 1980; Colbeck, 1982; Warren 1982), and a lower albedo increases the proportion of solar radiation absorbed at the ice-air interface, providing more energy for surface warming and melt. This positive feedback mechanism may have contributed to the tripling of glacier mass loss from this region between 2004–06 and 2007–09 (Gardner et al., 2011).'*

**Section 3.5 Comparison with the mean summer LST, added text (lines 395-397):**

> *'Owing to the positive feedback between albedo and surface temperature we would expect to observe strong increases in surface temperature where albedo declines were large. To investigate the relationship between temperature and albedo over the QEI, the 16-year mean summer LST record was compared with the 16-year mean summer BSA record (Sect. 2.3).'*

**Conclusion, added text (lines 476-485):**

*'Albedo declines increase the proportion of incoming solar radiation absorbed at the air-ice interface, and thus the energy available to drive melt, warming, and further surface albedo decline. Warmer temperatures, in turn, increase the rate of snow grain metamorphism which lowers the albedo. Air and surface temperatures affect the removal (timing and extent) of the seasonal snowpack which exposes lower albedo firn and/or glacier ice, while melting glacier ice releases impurities that further reduce its albedo. Given that surface temperature and albedo are inextricably linked, knowing where and when albedo changes are likely to occur in future is important for predicting future rates of mass loss from the QEI ice caps. Recent investigations of atmospheric circulation patterns over the QEI (e.g. Gardner and Sharp, 2007) focused on characterization of July temperature and atmospheric conditions, since July is usually the month when melt rates peak. Our results suggest, however, that changes occurring during the month of August are also important, especially as the length of the melt season continues to increase.'*

**RC**: The authors say that only the month of July shows statistically-significant albedo decrease over the study period. Thus, they should consider stating that the July albedo is decreasing over the study period instead of saying mean summer (JJA) albedo is decreasing.

**AR**: There appears to have been some misinterpretation of our results. No measurable changes in the June or August QEI-wide (regionally-averaged) BSA were observed owing to the large amount of spatial variability in the sign and magnitude of BSA change in these months. Measurable changes were observed in the mean summer (JJA) and mean July QEI-wide BSA. These changes, however, were not statistically significant at the $p < 0.05$ level. Although there were statistically significant (at the $p < 0.05$ confidence level) decreases in the mean summer (JJA) and/or July BSA at the pixel scale, this was not the case for either the mean summer (JJA) or July QEI-wide (spatially averaged) BSA change. As such, our conclusions and abstract remain unchanged. Text in the results section 3.3.1 (lines 306-308) has been revised for clarity.

**Revised text:**
> **Results section 3.3.1 lines 318-320**: *'Although the measured change in the QEI-wide mean summer BSA (average correlation coefficient of all pixels) was not statistically significant ($r = 0.31$, $p = 0.24$), BSA declines that were significant at the $p \leq 0.05$ level were observed at the pixel scale on all ice masses (Fig. 6a).'*

**Reviewer 1 specific comments:**

**RC 1** Title: consider removing "C6" from the title since it's not that different from C5 for the albedo and LST, and many people, before reading the paper, have no idea what C6 is. You have appropriately mentioned the fact that you've used C6 in the paper as needed. Also consider including LST in the title, especially if you decide to enhance the discussion about cause-and-effect between LST and albedo for the QEI.

**AR 1:** Reference to "C6" removed from the title. The title has also been modified to better reflect the content of the paper. Instead of 'Characterization of' we use 'Spatiotemporal variability of' to indicate that the paper investigates both the spatial and temporal patterns of glacier surface albedo in the Canadian High Arctic.

**Revised title:**
> *'Spatiotemporal variability of Canadian High Arctic glacier surface albedo from MODIS data, 2001-16'*

**RC 2:** Figure 1 caption: do you mean "Top-left inset" and "Bottom-right inset"?

**AR 2:** Bottom-left inset in Figure 1 removed and caption corrected.

**Revised Figure 1 caption:**

*'Figure 1: Glaciated regions of the Queen Elizabeth Islands. Background image: MODIS, 4 July 2011. Inset: red polygon shows location of Queen Elizabeth Islands, Arctic Canada.'*

**RC 3:** Figures 2,3,5,6,7&9: it should be mentioned in the caption that white (Figs 2 & 3) is not ice and brown is not ice (Figs 5,6,7, & 9).

**AR 3:** Mention of white areas (Fig. 2 & 3) and brown (Fig. 5, 6, 7, 9 (now Figure 8)) areas that are not ice has been added to the figure captions.

**Revised Figure captions:**

*'Figure 2: Mean summer clear-sky shortwave broadband black-sky albedo over the QEI ice caps. White areas outside of the ice caps indicate non-glaciated ice cover.*

*Figure 3: Mean summer clear-sky shortwave broadband albedo anomaly over the QEI ice caps relative to the 2001-16 mean. White areas outside of the ice caps indicate non-glaciated ice cover.*

*Figure 5: Linear rate of change (yr$^{-1}$) in (a) mean summer (JJA), (b) June, (c) July, and (b) August, clear-sky shortwave broadband black-sky albedo for 2001-16 over the QEI ice caps. Background image: MODIS, 4 July 2011. Brown indicates non-glaciated land cover.*

*Figure 6: p-value of the linear regression (Fig. 4) of (a) mean summer (JJA), (b) June, (c) July, and (d) August, clear-sky shortwave broadband black-sky albedo for the period 2001-16 over the QEI ice caps. Background image: MODIS, 4 July 2011. Brown indicates non-glaciated land cover.*

*Figure 7: Component scores for the first two Principal Components of the mean summer clear-sky BSA (Fig. 2) over the QEI ice caps. Background image: MODIS, 4 July 2011. Brown indicates non-glaciated land cover.*

*Figure 8: (a) Pearson Correlation Coefficient (r) and (b) p-value for linear regression of the 16 year BSA and LST record over the QEI ice caps. Background image: MODIS, 4 July 2011. Brown indicates non-glaciated land cover.'*

**RC 4:** Figure 8 – lower panel: I am confused about the point that you are trying to get across in this graph; please clarify.

**AR 4:** The lower panel of Figure 8 (which is now Figure 9) illustrates the correspondence between the Empirical Orthogonal Functions for the first and second Principal Components of the 16 year mean summer BSA record, respectively, and the mean summer NAO index. The graph presents the EOFs of PC1 and PC2 of both the mean summer (JJA) BSA record and the mean summer (JJA) NAO index for each year during the period 2001-16.

The original legend on the lower panel of Figure 8 (now Figure 9) did not include the NAO index. This may have been a source of confusion. The legend has been modified to include the NAO index (black line) and a reference to Section 3.4 where this graph is discussed, has been added.

**Modified Figure 9 with figure caption:**

[Figure]

[Figure]

*'Figure 9: (a) mean summer (JJA) clear-sky shortwave broadband black-sky albedo (left-hand axis) and the mean summer (JJA) NAO index (right-hand axis) for 2001-16. (b) Empirical Orthogonal Functions (EOF) for the First and Second Principal Components of the 16 year mean summer BSA record (left-hand axis), and the mean summer (JJA) NAO index (right-hand axis) for 2011-2016 (Sect. 3.4). '*

**RC 5:** Line 14: note that the range of years says 2008-2012 on line 486

**AR 5:** Corrected to read 2007-12 throughout.

**Revised text:**

> **Line 12-14:** *'Most of the decrease in BSA, which was greatest at lower elevations around the margins of the ice masses, occurred between 2007 and 2012 when mean summer BSA was anomalously low.'*

> **Line247-248:** *'The period 2001–16 is characterised by a six-year period of positive BSA anomalies (2001–06) followed by a six-year period of negative BSA anomalies (2007–12) (Table 1).'*

> **Line 250-252:** *'Negative BSA anomalies during the period 2007–12, which indicate a larger absorbed fraction of incoming shortwave radiation relative to the 16-year mean, coincide, and are consistent with, positive summer air and glacier surface temperature anomalies from 2007 to 2012 (Mortimer et al., 2016).'*

> **Line 278-279:** *'… in years when the mean summer QEI-wide BSA anomaly was strongly negative (e.g. 2001–04) (or positive (e.g. 2007–12))'*

**Line 418-420:** *'Between 2001 and 2016, in years with low (e.g. 2007–12) or high (e.g. 2001–04)) albedos, there was a persistent ridge (trough) in the 500 hPa geopotential height surface centered over the north and west of the QEI (Fig. 4; Sect. 3.1.1).'*

**Line 466-467:** *'Strong negative BSA anomalies from 2007–12 suggest that the bulk of the observed albedo decline occurred during this six-year period.'*

**RC 6** Line 60: MODIS has already been defined on lines 53 & 54

**AR 6:** Corrected.

**Revised line 60 (now line 61):**
*'Observations from the MODIS sensors, aboard the Terra (2000 to present) and Aqua (2002 to present) satellites,…'*

**RC 7** Line 76 & 78: Lutch should be spelled Lucht

**AR 7:** Misspelled reference corrected in **lines 76 & 78 (now lines 77 & 79)**

**Revised text:**
**Line 77:** *'…kernel-driven BRDF model (Wanner et al., 1997; Lucht et al., 2000; Schaaf et al., 2002, 201b).'*
**Line 79:** '*…is used to estimate the surface albedo (Strugnell and Lucht, 2001; Schaaf et al., 2002; Jin et al., 2003; Liu et al., 2009).'*

**RC 8** Line 99: perhaps the word "detectors" should be used instead of "instruments"?

**AR 8:** The word "instruments" replaced by "detectors".

Revised **Line 99 (now line 91-92:**
**'***The MODIS sensors are currently operating well beyond their expected [productive] six year lifetimes (Barnes et al, 1998; Justice 1998) and the detectors are degrading (Xiong et al., 2001).'*

**RC 9** Line 120-124: this discussion of MOD10A1 should be deleted; it is not relevant to this paper, and MOD10A1 is never mentioned again in the paper; it really isn't good validation for MCD43A3.

**AR 9:** Discussion of MOD10A1 has been deleted. Additionally, in response to an anonymous reviewer's comments, this section has been revised to be more informative regarding MODIS sensor design and capabilities.

**Revised text, lines 90-138:**

[revised manuscript text omitted]

**RC 10** Line 151: This is misleading because, in Hall et al (2008a), the pixels when the temperature was >2 deg C were removed since it's impossible to have a temperature greater than zero and still be ice. Hall et al. (2008a) did not remove pixels with errors >2 deg C that were below zero, if I am recalling correctly.

**RC 10.1:** Hall et al. (2008a) did not remove pixels with errors >2 deg C that were below zero, if I am recalling correctly.

**AR 10.1:** You are correct in pointing out that Hall et al. 2008a did not remove pixels with errors >2°C that were below 0°C. Hall et al. (2008a) developed melt-frequency maps and identified any pixels with a LST >0°C as experiencing melt. The inclusion of Hall et al. (2008a) at the end of this sentence is, as you say, incorrect and misleading. As such, the reference to Hall et al. (2008a) has been removed and only the reference to Mortimer et al. (2016) is included.

**Revised text, Line 151 (now line 177-179):**

*'Pixels for which the average LST error (QC_Day LST error flag) exceeded 2°C were removed from the analysis and any remaining pixels having a temperature >0°C were assigned a temperature of 0°C (Mortimer et al., 2016).'*

**RC 10.2:** …'it's impossible to have a temperature greater than zero and still be ice.':

**AR 10.2:** Although the temperature of pure ice and snow cannot exceed 0°C, many pixels are not comprised solely of pure ice and snow. Rock, dust, impurities and ponded water (for which the maximum temperature can exceed 0°C) can exist on the snow/ice surface. The presence of these materials within the 1km x 1km pixel can result in an LST >0°C. Mortimer et al. (2016) found that pixels having an LST >0°C were mainly located on outlet glaciers and near the ice-cap margins and inferred that such pixels probably contained a mixture of exposed rock, ice, and meltwater during the melt season. The presence of materials other than pure ice and snow within a pixel can, therefore, result in an LST greater than the maximum temperature of pure ice and snow. To mitigate this effect, ice-covered pixels with an LST >0°C were reassigned a value of 0°C.

**RC 11** Line 155: it should be stated that MOD11A2 Collection 6 data were downloaded

**AR 11** Corrected

**Revised Line 155 (now line 183-185):**
> *'MOD11A2 C6 data were downloaded from (https://lpdaac.usgs.gov/, accessed September–October 2017) and re-projected to a North America Albers Equal Area projection, WGS84 datum, 1 km resolution.'*

**RC 12** Line 235: I would say "opposite from" instead of "opposite to"

**AR 12** This text, as written, is not included in the revised manuscript.

**RC 13** Line 319: I think sections 3.3 and 4.1 could be shortened and made to be more concise.

**AR 13** Sections 3.3 (now section 3.4) and 4.1 (now section 4) have been shortened. Repetition in sections 3.3 and 4.1 has been removed. Sections 4.1 and 4.2 have been merged (now section 4).

**Revised Section 3.4, Lines 365-392:**

[revised manuscript text omitted]

**RC 14** Line 601: should read IEEE instead of IEE

**AR 14** Reference corrected (IEEE instead of IEE)

**Revised text:**

*'Salomon, J. G., Schaaf, C. B., Strahler, A. H., Gao, F., and Jin, Y.: Validation of the MODIS Bidirectional Reflectance Distribution Function and Albedo retrievals using combined observations from the Aqua and Terra platforms, IEEE Transactions on Geoscience and Remote Sensing, 44(6), 1555-1565, 2006.'*

---

## Author Comment (AC4) · 14 Nov 2017

[revised manuscript text omitted]

**Commented [m3]:** Rev. 2
RC 13: Page 2, line 38 – What is meant by 'accelerated release'? Suggestion to reword, clarify intent.
AR 13: Sentence reworded.

**Commented [m4]:** Rev. 2
RC 14: Page 2, lines 40-42 – The authors are correct to state fres snow would raise surface albedo. Suggestion to include QEI precipitation data quantifying the suggestion, or reference precipitation studies.
AR 14: Reference to Vincent et al. (2015) has been added. Vince al. (2015) found an increase in both total precipitation as well as increase in the ratio of snowfall to total precipitation over the Canadian Arctic during the period 1948-2012. These trends were calculated by gridding climate station data. In the QEI, there are three such stations (Resolute Bay, Eureka, Alert) and none are located over ice. Reference to Box et al. (2012) which found a second-order negative albedo feedback (positive correlation betw albedo and surface air temperature anomalies) in the accumulatio area of the nearby Greenland Ice Sheet has also been added to the of this sentence.

**Commented [m5]:** Rev. 2
RC 15: Page 2, line 49 – Suggestion to clarify for readers the sou of the temperature analysis used in referenced studies.
AR 15: clarification added.

and quantify the rate of albedo change across the QEI

**2 Data and methods**

**2.1 MCD43A3 data**

Observations from the MODIS , aboard the Terra (2000 to present) and Aqua (2002 to present) satellites (Barnes et al., 1998), are used to assess the spatial and temporal evolution of the surface albedo over the QEI glaciers and ice caps in summer (June-August). We use the MODIS/Terra and Aqua BRDF/Albedo Daily L3 Global – 500m Collection 06 (C6) product (MCD43A3, Schaaf and Wang (2015)), which provides both white-sky (bi-hemispherical reflectance under isotropic conditions) and black-sky (directional hemispherical reflectance) shortwave broadband surface albedo (Schaaf et al., 2011; https://www.umb.edu/spectralmass/terra_aqua_modis/v006, and references therein). MCD43A3 albedo is calculated daily for local solar noon  using atmospherically corrected surface reflectance measurements  made by sensors on both the Terra and Aqua satellites over a 16-day period that is centered on the ninth day of each 16-day moving window (https://www.umb.edu/spectralmass/terra_aqua_modis/v006). A semi-empirical Bidirectional Reflectance Distribution Function (BRDF) model, which describes the surface scattering/reflectance of a target as a function of illumination, is used to estimate surface albedo from directional surface reflectance information recorded by the MODIS sensors (Schaaf et al., 2002, 2011; Jin et al., 2003; Salomon et al., 2006). MCD43A3 white- and black-sky albedos are estimated from Level 2G-lite surface reflectances (MOD/MYD 09) for seven visible and near-infrared bands (spanning 0.4 to 2.4 µm) and three broad bands (shortwave (0.3-5.0 µm), visible (0.3-0.7 µm) and near infrared (0.7-5.0 µm)) in one of two ways.

If sufficient (>7) multi-date cloud-free observations with angular sampling sufficient to fully characterize the viewing/illumination geometry are acquired during a  16-day period, a high quality *full inversion* is run using a semi-empirical RossThick LiSparse Reciprocal (RTSLR) kernel-driven BRDF model (Wanner et al., 1997;  Lucht et al., 2000; Schaaf et al., 2002, 2011). If insufficient observations (<7) are available, then a lower quality *magnitude inversion*, which relies on a priori knowledge to scale an archetypal BRDF, is used to estimate the surface albedo (Strugnell and Lutch, 2001; Schaaf et al., 2002; Jin et al., 2003; Liu et al., 2009). Data quality flags, provided in the MCD43A2 data quality assessment product, indicate whether albedo values [for each pixel] were obtained using the *full* or *magnitude inversion*. Since this study aims to generate an initial assessment of the spatio-temporal variability of the surface albedo of glaciers and ice caps in the QEI with good spatial coverage, both *full* and *magnitude inversion* data are used. Although *magnitude inversions* produce lower quality albedo estimates than the *full inversion* method, previous work using the C5 data showed that the *magnitude*

**Commented [m6]: Rev. 1**
RC 6 Line 60: MODIS has already been defined on lines 53 & 54
AR 6: Corrected.

**Commented [m7]: Rev 1**
RC 7 Line 76 & 78: Lutch should be spelled Lucht
AR 7: Misspelled reference corrected.

*inversion* data provide a good representation of the seasonal and spatial patterns  of glacier surface albedo (Schaaf et al., 2011; Stroeve et al., 2013). To our knowledge, no recent research comparing the *magnitude* and *full inversion* retrievals over glaciers and ice caps has been published for the MCD43A3 C6 data. Comparison of the MODIS C5 and C6 *full inversion* albedo data from the Greenland Ice Sheet confirmed many of the broad spatial patterns in surface albedo identified in the C5 data, but the magnitude of the C6 albedo change was much smaller (Casey et al., 2017).

**2.1.1 MODIS sensor degradation**

The MODIS sensors are currently operating well beyond their expected [productive] six year lifetimes (Barnes et al, 1998; Justice 1998) and the detectors are degrading (Xiong et al., 2001). For both the MODIS Terra and Aqua sensors, instruments were calibrated pre-launch (radiometric, spatial, specular calibration) (Gunther et al., 1996). On-orbit calibration procedures were included to monitor the sensor degradation that is expected as the instruments are exposed to solar radiation (Gunther et al., 1996). For the reflective solar bands (0.41–2.2 µm) the onboard calibration system includes a solar diffuser (SD) calibrated using the solar diffuser stability monitor (SDSM) (Gunther et al., 1998). Lunar and Earth view observations (select desert sites) are also used to assess radiometric stability (Sun et al. 2003). Even so, long-term scan mirror and wavelength dependent degradation which are not sufficiently accounted for by the on-board calibrators (SD/SDMS) have been observed (Xiong et al., 2001; Lyapustin et al., 2014 and references therein). Calibration degradation effects, which are largely confined to the MODIS Terra sensor, are greatest in the blue band (B3) and decrease with increasing wavelength (Xiong and Barnes 2006). An anomaly in the SD door operation (3 May 2003) and a decision to leave the door permanently open, exposing the SD to additional solar radiation, resulted in degradation of the SD on the MODIS Terra sensor that was faster than had originally been anticipated, and than was observed for the MODIS Aqua sensor (Xiong et al., 2005). Differences in the response versus scan angle (RVS) for the two side mirrors were also characterized pre-launch (Barnes et al., 1998). The RVS is important because it describes the scan mirror's response to different angles of incidence (AOI, for each band, detector and mirror side) (Sun et al., 2014). However, for the MODIS Terra sensor, following an overheating incident during pre-launch calibration, the RVS was not re-characterized and the exact pre-launch RVS characteristics are not known (Pan et al., 2007; Sun et al., 2014 and references therein). These issues have resulted in the performance of the MODIS Terra sensor being poorer than that of the MODIS Aqua sensor.

As a normal part of the operational procedure, the MODIS Characterization Support Team (http:mcst.gsfc.nasa.gov) periodically updates the calibration algorithms and approaches, during which time the entire L1B record (calibrated top of the atmosphere radiances) is re-processed to reflect improved understanding and characterization of changes to the instruments. Even so, non-physical trends in MODIS Terra data products, that result from calibration drift, have been observed and are well documented (e.g. Xiong et al., 2001; Xiong and Barnes, 2006; Franz et al., 2008; Kwiatkowski et al., 2008; Wang et al., 2012; Lyapustin et al., 2014; and references therein). The latest revision occurred with the C6 data and includes on-orbit calibration procedures to mitigate long-term calibration drift, particularly at the shorter wavelengths (Wenny et al., 2010; Toller et al.,

**Commented [m8]:** Rev. 2
RC 20: The paragraph (Page 4, Paragraph 3) could be rewritten to more informative and clear regarding MODIS sensor design and capabilities.
AR 20: MODIS Characterization Support Team literature has been reviewed and this section has been re-written to be more informa

**Commented [m9]:** Rev. 2
RC 16: Page 4, line 98 – Delete 'some of'.
AR 16: Deleted 'some of'

**Commented [m10]:** Rev. 1
RC 8 Line 99: perhaps the word "detectors" should be used instead of "instruments"??
AR 8: The word "instruments" replaced by "detectors".

**Commented [m11]:** Rev. 2
RC 17: Page 4, line 102 – Lyapustin et al. was not the first to report on Terra's band degradation. Recommendation to additionally read and cite early / appropriate work, e.g. Xiong et al., 2001, Degradation of MODIS optics and its reflective solar bands calibration, doi: 10.1117/12.450646
AR 17: References to earlier studies have been added.

**Commented [m12]:** R2: Page 4, line 102 – Lyapustin et al. was not the first to report on Terra's band degradation. Recommendation to additionally read and cite early / appropriate work, e.g. Xiong et al., 2001, Degradation of MODIS optics and its reflective solar bands calibration, doi: 10.1117/12.450646
Xiong and Barnes, 2006, An overview of MODIS radiometric calibration and characterization, doi: 10.1007/s00376-006-0008-4
Sun et al., 2014 Time-dependent response versus scan angle for MODIS reflective solar bands, IEEE TGRS, doi: 10.1109/TGRS.2013.2271448

[revised manuscript text omitted]

**Commented [m13]: Rev. 2**
RC 18: Page 4, paragraph 3. Suggestion for authors to reread literature on MODIS sensor calibration, degradation and capabili Line 110, more correct to state that C6 did improve radiance measurements from launch to ~2013. It remains to be assessed ho accurate and reliable MODIS C6 data will be moving forward fro C6 implementation (~2013 to present). Line 110: Recommendat to check the MODIS Characterization Support Team literature https://mcst.gsfc.nasa.gov/publications?f%5Btype%5D=102 .
AC 18: Literature has been reviewed and note about improved radiances up until ~2013 has been added.

**Commented [m14]: Rev. 2**
RC 19: For lines 113-114, it is not that the sensor is capable of identifying trends greater than 0.01, so much as +/- 0.01 is the lim of MODIS sensor accuracy and precision. The paragraph could b rewritten to be more informative and clear regarding MODIS sen design and capabilities.
AC 19: This paragraph has been re-written to be more informativ and accurate regarding MODIS sensor design and capabilities. T text 'is capable of identifying trends greater than 0.01' has been removed.

**Commented [m15]: Rev. 2**
RC 16: Page 3, lines 89-91 – Suggestion to edit sentences. As written, one could glean that it is difficult to discriminate between surface snow and ice vs cloud spectral visible-thermal infrared response. This is not always difficult to do spectrally. If the autho intend to discuss cloud remote sensing only, please clarify this. A more appropriate reference than Hall 2008a, may be Hall et al., 2 MODIS snow-cover products, Remote Sensing of Environment, 3 181-194.
AR 16: text moved and re-written for clarity. Reference to Hall 2 has been added. The issue of concern here is the absence of cloud which can potentially introduce additional variability in the datas that is not representative of physical changes on the surface.

term ground measurements of glacier surface albedo in the QEI, we made the assumption that this is also the case in that region.

160 The MODIS sensors are currently operating well beyond their expected [productive] six year lifetimes and some of the instruments are degrading (Wang et al., 2012; Toller et al., 2013). Systematic (decreasing) temporal trends are present in measurements in the visible and NIR (bands 1-7) of the MODIS C5 data (Lyapustin et al., 2014). Calibration degradation effects, which are largely confined to the Terra sensor, are greatest in the blue band (B3) and decrease with increasing wavelength (Lyapustin et al., 2014). Over time, un-corrected sensor degradation gives rise to decreasing measured surface

165 radiances, which may result in apparent MODIS-derived albedo declines that differ from the true physical change. Contrary to the findings of earlier work that identified strong albedo declines over the dry snow zone of the Greenland Ice Sheet from 2001-2013 using the C5 data (e.g. Stroeve et al., 2005, 2013; Box et al., 2012; Alexander et al., 2014), a recent investigation concluded that much of the observed decline in broadband albedo from Terra only data resulted from sensor degradation (Polashenski et al., 2015).

170 Long-term drifts in sensor calibration are addressed in the C6 data used in this study. A new calibration technique that uses in-orbit data from pseudo time-invariant desert site targets is in the calculation of C6 surface reflectances (Toller et al., 2013). This vicarious calibration approach has improved the precision of C6 reflectances, and it also mitigates long-term sensor drift, particularly at the shorter wavelengths. The C6 post-calibration residual error is on the order of several tenths of one percent for top of the atmosphere (TOA) reflectance (Lyapustin et al., 2014), but larger discrepancies between calibrated Aqua and

175 Terra measurements have been observed in the most recent data (post 2014) (Casey et al., 2017). MODIS C6 data have been shown to be capable of identifying trends in surface albedo >0.01 decade$^{-1}$ (Lyapustin et al., 2014). A recent analysis of summer (June-August) ice surface albedo changes from MODIS C6 data identified statistically significant declines in surface albedo over the wet snow zone of the Greenland Ice Sheet during the period 2001-2016. For the most part, these decreases in surface albedo are thought to be physically real (Casey et al., 2017).

180 There are no long-term, spatially distributed, in situ albedo records from the glaciers and ice caps in the QEI, so a comparison between the MCD43A3 records and ground observations is not possible, and this is a limitation of this study. Ground-truthing of the MOD10A1 C6 albedo product over the Greenland Ice Sheet, immediately adjacent to the QEI, has been undertaken using in situ measurements from the Greenland Climate Network (GC-Net) and the Programme for Monitoring of the Greenland Ice Sheet (PROMICE) (Box et al., 2017). This study found the MOD10A1 C6 data to be a reasonable representation

185 of the true surface albedo. In the absence of quality spatially distributed field measurements of surface albedo from our study area, we assume that this is also the case on the QEI ice caps.

Commented [m16]: Rev. 2
RC 2: "In the data and methods section (Page 4, line 122) MOD10A1 data was mentioned "to be a reasonable representation of the true surface albedo". Does this refer to analysis conducted by the authors for this manuscript or to Box et al., 2017? Please clarify. MOD10A1 data was not inspected by the authors for this manuscript, why not?"
AC 2: The reference to the MOD10A1 data has been removed and the methods section detailing MODIS sensor calibration issues has been revised.

Commented [m17]: Rev 1
RC 9 Line 120-124: this discussion of MOD10A1 should be deleted; it is not relevant to this paper, and MOD10A1 is never mentioned again in the paper; it really isn't good validation for MCD43A3
AR 9: Discussion of MOD10A1 has been deleted. Additionally, in response to an anonymous reviewer's comments, this section has been revised to be more informative regarding MODIS sensor degradation and capabilities.

**2.1 1.2 MCD43A3 data processing**

Summer (1-2 June (day 152) to 30-31 August (day 243)) MODIS MCD43A3 and MCD43A2 Collection 6 (Schaaf and Wang, 2015) data for MODIS tiles h17v00, h16v00, h16v01, and h15v01 for the period 2001-2016 were obtained from the NASA/USGS Land Processes Distributed Active Archive Center (Schaaf and Wang (2015), http://lpdaac.usgs.gov/ accessed November 2016). Daytime clear-sky white- and black-sky shortwave broadband and visible albedo data (MCD43A3) and accompanying quality assessment information (MCD43A2) were extracted from the hierarchical data format files and re-projected from the standard MODIS sinusoidal projection to a North America Albers Equal Area projection, WGS84 datum, 500 m resolution, using the MODIS re-projection tool version 4.1 (https://lpdaac.usgs.gov/tools/modis_reprojection_tool). The maximum summer (June-August) solar zenith angle over our study area (74°) was below the product's stated accuracy (<75°, Vermote et al., 2011; Wang et al., 2012), so no additional filtering was performed to remove data with high solar zenith angles. The white- and black sky albedos (representing completely diffuse and completely direct illumination, respectively) represent extreme estimates of the actual (blue-sky) bi-hemispheric surface albedo. To avoid redundancy, only results for the black-sky albedo (BSA) (which are fully consistent with those obtained using the white-sky albedo (WSA)) are presented here. The BSA was selected because our analysis focuses on albedo retrieved under clear-sky conditions. This approach is consistent with previous work using MCD43A3 data (e.g. Alexander et al., 2014; Tedesco et al., 2016; Casey et al., 2017). Unless otherwise specified, henceforth BSA refers to the shortwave broadband black-sky albedo.

**2.2 MODIS LST (MOD11A2)**

Warmer surface temperatures increase the rates grain metamorphism and snowmelt, resulting in larger snow grains which have a lower albedo than those of fresh snow (Wiscombe and Warren 1980; Colbeck 1982) (Sect. 1.0). Air and surface temperatures also affect the timing of removal of the seasonal snowpack, which exposes lower albedo firn or glacier ice. Additionally, the melt of glacier ice releases impurities that have a low albedo and thus change the surface albedo of both ice and snow (e.g. Clarke and Noone 1985; Doherty et al., 2010; Sect. 1.0). As such, analysis of the glacier surface temperature and comparison of these data with the albedo record is included to help understand the observed spatiotemporal patterns of glacier albedo change in the QEI.

We use tThe Eight-Day L3 Global Land Surface Temperature and Emissivity product (MOD11A2) C5C6, which has been found to be a reasonable proxy for the duration and/or intensity of summer melting in the QEI (Sharp et al., 2011; Mortimer et al., 2016), was used to investigate the relationship between glacier surface temperature and albedo. Only the 'daytime' LST data, generated with the day/night algorithm of Wan and Li (1997), is evaluated here. This is consistent with previous work in this region (e.g. Sharp et al., 2011, Mortimer et al. 2016). MOD11A2 daytime and night-time LSTs are computed from MODIS channels 31 (11 μm) and 32 (12 μm) using a split-window technique and all available daytime clear-sky scenes from the Terra satellite for sequential eight day periods (Wan et al., 2002). These data have a spatial resolution of 1 km and nominal product accuracy of ± 1°C but the accuracy over snow and ice surfaces can be as low as ±2°C (Hall et al., 2008z b; Koenig and Hall,

**Commented [m18]:** Rev. 2
**RC 21:** Page 5, line 129 – Suggestion to move citation to correct location in the sentence, i.e. Schaaf and Wang 2015 reference sho immediately follow MCD43 product mention.
**AR:** citation moved.

**Commented [m19]:** Rev. 2
**RC 22:** Page 5, line 141 – Awkward as written, suggestion to rev to clarify further use of BSA term e.g. 'henceforth BSA refers to black sky albedo MODIS shortwave broadband data.'
**AR 22:** Line no longer included in manuscript.

**Commented [CM20]:** Rev 1.
**RC:** Though they point out that albedo and LST are negatively correlated for the QEI, they do not discuss that relationship as ca and-effect in either the Abstract or the Conclusion. The relationsh between albedo and LST is discussed in Section 4.2 but I would l liked that discussion to have been better integrated throughout th paper.

**Commented [m21]:** Rev. 2
**RC 4:** "Section 2.2 discusses MODIS LST data. Why was Colled 5 MOD11A2 data used? (Page 5, lines 155-157) Communication with the MODIS data distributor, LP DAAC, revealed that MOD11A2 and MYD11A2 Collection 6 data have been available since mid-2015. Additionally, with the discussion of the Terra se degradation, it seems short-sighted to use Terra data only. Why w Terra data used? Was Aqua data used (i.e. MYD11A2)? If not, w not? Recommendation to add MYD11A2 data analysis."
**AR 4:** The manuscript has been revised and now uses the C6 MOD11A2 data instead of the C5 product. Terra data was used because of the time period under investigation with Terra data be since 2000 and Aqua data only being available since 2002. The T data allowed comparison with the albedo record for the full time period (2001-16).

220 2010). The LST analyses presented here are an update of those presented by Mortimer et al. (2016) which used the MODIS C5 data. The two analyses also differ in the time period used (we use 2001-16 instead of 2000-15 to coincide with the BSA analysis). Pixels for which the average LST error (QC_Day LST error flag) exceeded 2°C were removed from the analysis and any remaining pixels having a temperature >0°C were assigned a temperature of 0°C (e.g. Hall et al., 2008a; Mortimer et al., 2016). Uncertainties in trends derived from the MOD11A2 LST data in the MOD11A2 LSTs arise mainly from cloud

225 contamination (Box et al., 2012; Hall et al., 2012) and the removal of observations for periods when clouds are detected (Ackerman et al., 1998; Hall et al., 2008ba). Variability in the number of clear-sky days within each observation period and from one year to the next was not found to introduce significant variability in the MODIS-derived LST relative to the true near-surface air temperature in the QEI (see Mortimer et al., 2016). MOD11A2 C6 data were downloaded from (https://lpdaac.usgs.gov/, accessed September 2014 - October 2015 and June 2017) and re-projected to a North America Albers

230 Equal Area projection, WGS84 datum, 1 km resolution.

**2.3 Mean summer BSA and LST**

Annual precipitation in the QEI is low (<400 mm yr$^{-1}$) and varies little from one year to the next. In contrast, the annual temperature range is large (> 40°C) (Braithwaite, 2005). Inter-annual variability in QEI annual mass balance is dominated by changes in the summer mass balance (Koerner, 2005), which, in turn, is strongly correlated with summer air temperature

235 (Sharp et al., 2011). Spatial and temporal patterns in BSA and LST were, therefore, evaluated for the summer months (June-August). For each year during the 2001-2016 period, mean summer (JJA) BSA was calculated for pixels having at least 10 BSA observations in each month (June, July, August) and at least 45 of a possible 92 observations during the JJA period. These monthly thresholds ensure both an even distribution of BSA data throughout the summer season and consistency between different years. Mean summer LST was calculated following the methods of Mortimer et al. (2016) where the mean

240 summer LST is calculated for pixels having at least 7 of a possible 12 observations between 1-2 June (day 153) and 28-29 August (day 241).

The mMean summer (JJA) BSA and LST and the mean monthly (June, July, August) BSA, as well as the BSA and LST anomalies, anomalies were calculated on a pixel-by-pixel basis relative to the 2001-2016 mean for pixels having mean summer observations in 11 or more years. This These constituted ~87% and ~9398% of possible BSA and LST pixels, respectively.

245 Long-term rates of change in BSA and LST over the period 2001-2016 were determined by linear regression between the 16 year records of mean summer LST and / BSA and time. Consistent with the BSA and LST anomalies, regressions were computed on a pixel-by-pixel basis for all pixels having mean summer observations for 11 or more years. Following Casey et al. (2017), BSA trends between -0.001 yr$^{-1}$ and +0.001 yr$^{-1}$ are considered to be negligible. Negligible trends were defined by Casey et al. (2017) on the basis of the magnitude of the residual calibration uncertainties in the C6 data (on the order of several

250 tenths of one percent in TOA reflectance (Lyaputsin et al., 2014)) over pseudo-invariant desert sites (Sect. 2.1.1). albedo trends below the detection limit of the C6 data (0.01 decade$^{-1}$ (Lyapustin et al., 2014)) are considered to be negligible.

**Commented [m22]: Rev. 2**
RC 11: Recommendation for the authors to clearly state the additional QEI LST analysis provided in this manuscript as compared to Mortimer et al., 2016.
AR 11: Following your recommendations, the manuscript has been revised and now includes the MOD11A2 C6 data instead of the C5 data. Text has been added to the Methods (Section 2.2) to clearly state the differences in the QEI LST analysis provided in this manuscript compared to Mortimer et al., 2016.

**Commented [m23]: Rev. 1**
RC 10.1: Hall et al. (2008a) did not remove pixels with errors >2°C that were below zero, if I am recalling correctly.
AR 10.1: You are correct in pointing out that Hall et al. 2008a did not remove pixels with errors >2°C that were below 0°C. Hall et al. (2008a) developed melt-frequency maps and identified any pixels with a LST >0°C as experiencing melt. The inclusion of Hall et al. (2008a) at the end of this sentence is, as you say, incorrect and misleading. As such, the reference to Hall et al. 2(008a) has been removed and only the reference to Mortimer et al. (2016) is included.
RC 10.2: …'it's impossible to have a temperature greater than zero and still be ice.':
AR 10.2: Although the temperature of pure ice and snow cannot exceed 0°C, many pixels are not comprised solely of pure ice and snow. Rock, dust, impurities and ponded water (for which the maximum temperature can exceed 0°C) can exist on the snow/ice surface. The presence of these materials within the 1km x 1km pixel can result in an LST >0°C. Mortimer et al. (2016) found that pixels having an LST >0°C were mainly located on outlet glaciers and on the ice-cap margins and inferred that such pixels probably contain a mixture of exposed rock, ice, and meltwater during the melt season. The presence of materials other than pure ice and snow within a pixel can, therefore, result in an LST greater than maximum temperature of pure ice and snow. To mitigate this effect, ice-covered pixels with an LST >0°C were reassigned a value of 0°C.

**Commented [m24]: Rev. 2**
RC 23: Page 5, lines 151-152 – The authors seem to generalize in that "uncertainties in the MOD11A2 LSTs arise mainly from cloud contamination". Suggestion to reread relevant literature and present accurately. Does the sentence refer to over snow only?
AR 23: This sentence referred to uncertainties in the trends derived from the MOD11A2 data and not in the MOD11A2 data itself. The text has been revised for clarity.

**Commented [m25]: Rev. 1**
RC 11 Line 155: it should be stated that MOD11A2 Collection 6 were downloaded
AR 11: C6 added.

**Commented [m26]: Rev. 2**
RC 24: Page 6, line 159 – Is there a reference from precipitation records/data in QEI? i.e. what station, record or data is 400 mm/yr derived from?
AR 24: The reference for the 400 mm/yr should be Braithwaite, 2005. This reference, included at the end of the following sentence, refers to both the temperature and precipitation ranges. Omitted reference now included.

To explore whether there were data contained any other spatial patterns that differed from the long-term (linear) trend, a Principal Components Analysis of the 16 year mean summer BSA record was performed using data from all pixels with mean summer BSA observations in every year (50% of pixels).

To investigate the spatial pattern of the relationship between surface temperature (LST) and albedo (BSA), linear correlations between the 16 year LST and BSA records were computed. The MCD43A3 C6 albedo data are produced daily whereas the MOD11A2 LST data are produced only every eight days. For this direct comparison between the LST and BSA data, eight day BSA averages were computed from the daily data for the same eight day periods as the MOD11A2 LST product, and resampled to a 1 km spatial resolution (nearest neighbour resampling). For each year, mean summer BSAs were computed from these eight day averages for all pixels having at least seven 7 of a possible twelve 12 observations, consistent with the computation of mean summer LST. (Sect. 2.2). The difference between the mean summer BSA values derived from these 8-day averages and those computed from the daily data (0.008) is within the uncertainty of MODIS reflectance products (0.05 for solar zenith angle <75°; Vermote et al., (2011)). Linear correlations between the 16 year BSA (eight day averaged) and LST records were then computed on a pixel-by-pixel basis for all pixels having LST and BSA observations in all years (~8046% of all possible pixels).

To ensure that only data for glaciated surfaces were retained, all BSA and LST outputs used in this analysis were clipped to the Randolph Glacier Inventory v3.2 region 32 (Arctic Canada North) reference polygons (Arendt et al., 20132012; Pfeffer et al., 2014). Surface elevations were obtained from the Canadian Digital Elevation Dataset (CDED) edition 3.0, scale 1:50 k, re-sampled to a 500 m resolution.

**3 Results**

**3.1 Mean summer albedo**

Annual maps of the mean summer clear-sky broadband shortwave black-sky MCD43A3 albedo for all glacier-covered surfaces in the QEI for the 2001-2016 period are presented in Fig. ure 2. The QEI-wide mean summer BSA, averaged across all 16 years, was 0.550 ± 0.115 (mean ± 1 standard deviation; Table 1). The lowest QEI-wide mean summer BSA (0.539 ± 0.127) was recorded in 2011 while the highest (0.668 ± 0.089) was recorded in 2013 (Table 1).

In general, mean summer BSA is lower around the margins of the ice masses, where glacier ice is exposed in the summer, than it is in the higher elevation interior regions where snow or firn are exposed year-round (Fig. 2). During years when the QEI-wide mean summer BSA was low (e.g. 2011), we observed a broad zone of low albedo values (< 0.4) around the margins of the major ice masses (Fig. 2). Conversely, in years when the mean summer BSA was high (e.g. 2013), this zone was much less obvious. High data dropout at high elevations on Axel Heiberg Island, and over the summit of the Devon Ice Cap in 2014 and 2006 (Table S1), may have produced a negative albedo bias for these regions, since the albedo is typically greater at higher

Commented [m27]: Rev. 2
RC 8: Include references to Tables, methods as appropriate.
AR 8: Additional references to Tables, Figures, and methods have been added throughout.

Commented [m28]: Rev. 2
RC 5: "Reorganization of content is advised toward readability. There are several cases where content is difficult to follow. Exam Results Section, 3.1 Suggestion to add sentences clarifying and further detailing results. As written, parts of the section are terse non-intuitive."
AR 5: Section 3.1 has been re-ordered and additional text has be added for readability.

Commented [m29]: Rev. 2
RC 25: Page 7, line 195 – Suggestion to include MCD43A3 for clarity and completeness.
AR 25: Mention of MCD43A3 included for clarity and complete as suggested.

elevations. Aggregating the 2001-16 average mean summer BSA into 50 m elevation bins, we observed a linear rate of BSA increase with elevation (0.0085 per 50 m elevation bin, r² = 0.99).

In addition to the mean summer (JJA) BSA, the monthly mean BSA values for June, July, and August, were also investigated
285 (Sect. 2.3) (Table 2, Fig. S1-S3). July had the lowest 16-year monthly mean BSA (0.551 ± 0.131), followed closely by August (0.579 ± 0.127) (Table 2, ). In each year during the 2001-16 period, the highest summer monthly BSA was always recorded in June while the lowest monthly BSA was recorded in either July or August. The lowest monthly mean BSA values for June and August were recorded in 2011, while the lowest  July, mean BSA occurred in 2012. The highest monthly mean BSA for both June and July occurred in 2013; for August, it occurred in 2003.
290

~~In general, mean summer BSA is lower around the margins of the ice masses than in the higher elevation interior regions (Fig. 2). Aggregating the 2001-2016 average mean summer BSA into 50 m elevation bins, we observed a linear rate of BSA increase with elevation (0.0085 per 50 m elevation bin, r² = 0.99). During years when the QEI-wide mean summer BSA was low (e.g. 2011), we observed a broad zone of low albedo values (< 0.4) around the margins of the major ice masses (Fig. 2). Conversely,~~
295

**3.2 Albedo anomalies**

**3.2.1 Mean summer (JJA) BSA anomalies**

300 The mean summer BSA anomaly, relative to the 2001-16 mean, are presented in Figure 3 and Table 1. For consistency with the regression analysis (Sect. 3.3), BSA anomalies were only computed for pixels having mean summer BSA observations in 11 or more of the 16 years (Sect. 2.3). The period 2001-16 is characterised by a six-year period of positive BSA anomalies (2001-06) followed by a six year period of negative BSA anomalies (2007-12) (Table 1).
305  Positive BSA anomalies were also observed in 2013 (+0.060) and 2014 (+0.015), while 2015 (-0.022) and 2016 (-0.005) saw a return to negative anomalies (Table 1).  Negative BSA anomalies during 2007-12, which indicate a larger absorbed fraction of incoming shortwave radiation relative to the 16-year mean, coincide, and are consistent with, positive summer air and glacier surface
310 temperature anomalies in the QEI from 2007 to 2015 (Mortimer et al., 2016). Higher temperatures increase the rate of snow grain metamorphism which lowers the surface albedo (Wiscombe and Warren, 1980; Colbeck, 1982; Warren 1982), and a lower albedo increases the proportion of solar radiation absorbed at the ice-air interface, providing more energy for surface

**Commented [m30]: Rev. 2**
RC 7: For example, readers may appreciate a sentence stating that in addition to the JJA averages, monthly mean albedo was also calculated (Table 2, see Section 2.3).
AR 7: Section 3.1 (presented above) has been re-ordered to make it easier to follow and a sentence (Line 209) regarding the monthly BSA mean albedo with reference to the appropriate Table, Figure and methods section, has been added.

**Commented [m31]: Rev. 2**
RC 6: Lines 196-197 Why does Table 1 2001-2016 average differ from manuscript stated average (i.e. Table 1 states 0.599, manuscript states 0.550)?
AR 6: The value of 0.599 in Table 1 refers to the mean summer BSA. The value of 0.551 on line 196-107 refers to the July mean BSA which is 0.551 ± 0.0131 and is presented in Table 2.

**Commented [m32]: Rev. 2**
RC 10: Two specific areas of speculation are: (1) Results Section 3.1, line 203, 'may indicate early onset of snowfall that fall' and (2) Results Section 3.2, lines 290-292 snowfall patterns may be changing in QEI. Quantified data based statements are preferred over speculation where possible.
AC 10: Again, while we agree that quantitative statements are preferred over speculation the absence of in-situ precipitation limits our ability to do this. The specific areas of concern noted in RC 10 have been removed from the manuscript.

[revised manuscript text omitted]

**Commented [m33]: Rev. 2**
RC 26: Page 9, line 272 – Note that it is not only the calibration accuracy that limits the capability to measure trends, but also the sensor design. Please add reference to sensor capabilities (e.g. Ju et al., 1998, doi: 10.1109/36.701075 and/or similar on MODIS instrument design and post-launch capabilities, see https://mcst.gsfc.nasa.gov/publications?f%5Btype%5D=102 ).
AR 26: Sentence revised for accuracy in both the methods and re sections. Additional references to the appropriate methods section and literature have been included.

[revised manuscript text omitted]

**Commented [m35]: Rev. 2**
**RC 27:** Page 12, lines 365-366, Suggestion to reword sentence, avoiding use of "positive (negative)" words side by side.
**AR 27:** We find this to be an efficient way of writing. No change made.

[revised manuscript text omitted]

**Commented [CM37]:** Rev. 1
**RC**: Though they point out that albedo and LST are negatively correlated for the QEI, they do not discuss that relationship as cause-and-effect in either the Abstract or the Conclusion. The relationship between albedo and LST is discussed in Section 4.2 but I would liked that discussion to have been better integrated throughout the paper.
**AC**: Additional text regarding the relationship between temperature and albedo has been included in the conclusion.

[revised manuscript text omitted]

**Commented [m41]: Rev. 2**
**RC 32:** Figure 1 image – In some formats, it is difficult to differentiate the thematic colors used in the figure. Consider if the may be other colors to use. For reader friendliness, it may also he move the legend and increase the font size of the legend text. Als consider adding a label to the 8 regions. This may help readers in interpretation of Supplemental Table S2
**AR 32:** Figure 1 revised. Bottom-right inset no longer included because regional BSA values are no longer referred to in text. La for the major ice masses have been included on the main map. Supplemental Table 2 has been removed.

**Commented [m42]: Rev. 2**
**RC 31:** Figure 1 caption – Do the authors intend to reference Moderate Resolution Imaging Spectroradiometer instead of "Moderate Resolution Spectroradiometer"?
**AR 31:** Figure 1 caption corrected to read Moderate

**Commented [m43]: Rev. 1**
**RC 2:** Figure 1 caption: do you mean "Top-left inset" and "Botto right inset"?
**AR 2:** Bottom-left inset in Figure 1 removed and caption correcte

[Figure]

1000

**Figure 2: Mean summer clear-sky shortwave broadband black-sky albedo**  **over the QEI ice caps. White areas outside of the ice caps indicate non-glaciated ice cover.**

1005

[Figure]

**Figure 3: Mean summer clear-sky shortwave broadband albedo anomaly**  over **the QEI** ice caps **relative to the 2001-****16 mean.** White areas outside of the ice caps indicate non-glaciated ice cover.

[Figure]

[Figure]

**Figure 4: Mean summer (JJA) composite NCEP/NCAR Reanalysis 500 hPa geopotential height anomaly for (a) a period of large negative BSA anomalies (2001-04) and (b) a period of large positive BSA anomalies (2007-12). Source: https://www.esrl.noaa.gov/psd.**

**Commented [m44]: Rev. 2**
**RC 33:** Figure 4 – Figure text, misspelled word 'anomaly' in two locations, please correct.
**AR 33:** Figure 4 correct for two erroneous spellings of 'anomaly'

[Figure]

**Figure 5:** Linear rate of change (yr⁻¹) in (a) mean summer (JJA), (b) June, (c) July, and (b) August, clear-sky shortwave broadband black-sky albedo for 2001- over the QEI ice caps. Background image: MODIS 4 July 2011rown indicates non-glaciated land cover.

Commented [m45]: Rev. 1
RC 3: Figures 2,3,5,6,7&9: it should be mentioned in the caption
white (Figs 2 & 3) is not ice and brown is not ice (Figs 5,6,7,&9)
AR 3: Mention of brown (Fig. 5, 6, 7, 9) areas that are not ice ha
been added to the figure captions.

[Figure]

**Figure 6: p-value of the linear regression (Fig.4) of (a) mean summer (JJA), (b) June, (c) July, and () August, clear-sky shortwave broadband black-sky albedo for the period 2001-16  over the QEI ice caps. Background image: MODIS 4 July 2011, Brown indicates non-glaciated land cover.**

[Figure]

1025

Figure 7: QEI (a) 16 year average mean summer land surface temperature (°C) and (b) linear rate of change of mean summer LST (°C yr⁻¹) for the period 2001-16 over the QEI ice caps. Background image: MODIS 4 July 2011., bBrown indicates non-glaciated land cover.

Commented [m46]: Rev. 2
RC 35: Figure S7 – Example of important, interesting QEI LST content for the authors to consider moving from the supplementa material to the main manuscript.
AR 35: Figure S7 moved from supplemental material to the main and is now Figure 7.

[Figure]

[Figure]

**Figure 78: Component scores for the first two Principal Components of the mean summer clear-sky BSA (Fig. 2)**  over **the QEI** **ice caps****. Background image: MODIS 4 July 2011, **Bb**rown indicates non-glaciated land cover.**

1035

[Figure]

[Figure]

[Figure]

[Figure]

**Figure 89: (a) mean summer (JJA) clear-sky shortwave broadband black-sky albedo (left-hand axis) and the mean summer (JJA) NAO index (right-hand axis) for 2001-2016. (b) Empirical Orthogonal Functions (EOF) for the First and Second Principal Components of the 16 year mean summer BSA record (left-hand axis), and the mean summer (JJA) NAO index (right-hand axis) for 2011-2016 (Sect. 3.4).**

Commented [m47]: Rev 1.
RC 4: Figure 8 – lower panel: I am confused about the point that are trying to get across in this graph; please clarify.
AR 4: The lower panel of Figure 8 (which is now Figure 9) illust the correspondence between the Empirical Orthogonal Functions the first and second Principal Components of the 16 year mean summer BSA record, respectively, and the mean summer NAO in The graph presents the EOF of PC1 and PC2 of the mean summe (JJA) BSA record as well as the mean summer (JJA) NAO index each year during the period 2001-2016.
The original legend on the lower panel of Figure 8 (now Figure 9 not include the NAO index. This may have been a source of confusion. For clarification, the legend has been modified to incl the NAO index (black line) and a reference to Section 3.3 where graph is discussed, has been added.

[Figure]

[Figure]

1045    **Figure 910: (a) Pearson Correlation Coefficient (r) and (b) p-value for linear regression of the 16 year BSA and LST record.**  Background **image**: MODIS 4 July 2011 **Bb**rown indicates non-glaciated land cover.